# LayerAct: Advanced activation mechanism utilizing layer-direction normalization for CNNs with BatchNorm

## Abstract

In this work, we propose a novel activation mechanism aimed at establishing layer-level activation (LayerAct) functions for CNNs with BatchNorm. These functions are designed to be more noise-robust compared to existing element-level activation functions by reducing the layer-level fluctuation of the activation outputs due to shift in inputs. Moreover, the LayerAct functions achieve this noise-robustness independent of the activation's saturation state, which limits the activation output space and complicates efficient training. We present an analysis and experiments demonstrating that LayerAct functions exhibit superior noise-robustness compared to element-level activation functions, and empirically show that these functions have a zero-like mean activation. Experimental results with three clean and three out-of-distribution benchmark datasets for image classification tasks show that LayerAct functions excel in handling noisy datasets, outperforming element-level activation functions, while the performance on clean datasets is also superior in most cases.

## 1 Introduction

Convolutional neural networks (CNNs) have been used as a primary architecture for computer vision tasks since the development of AlexNet (Bai et al., 2021; Wang et al., 2022; Wightman et al., 2021). Batch normalization (BatchNorm, Ioffe & Szegedy (2015)) has significantly contributed to the success of CNNs with its numerous benefits during training, such as ensuring stability, facilitating more efficient parameter optimization, and addressing the channel collapse problem of activation (Daneshmand et al., 2020; Ioffe & Szegedy, 2015; Santurkar et al., 2018). To build upon the considerable success of CNNs with BatchNorm, and in an effort to further advance the architecture, we pay close attention to the activation mechanism within network layers. Specifically, we propose a novel layer-level activation (LayerAct) mechanism, along with two associated functions. While maintaining the batch-direction normalization methods, which are effective and prevalent in CNN-based networks, the LayerAct mechanism can provide the benefits of layer-direction normalization. Specifically this mechanism offers two main advantages: i) addressing the trade-off issue between two significant properties of activation, and ii) improving the noise-robustness of activations by reducing the variance of noise-robustness across samples.

The majority of existing activation functions operate on an element (i.e. the activation functions operate on each element of a layer separately), exhibit limitations due to their element-level activation mechanism. Firstly, a trade-off exists between one-side saturation (limiting negative output space) and allowing negative outputs for zero-like activation mean. Activation functions that saturate only on one side are expected to have more informative propagation during the backward pass compared to those that saturate on both the positive and negative sides, by allowing for larger derivatives. (Clevert et al., 2016; Glorot et al., 2011) To achieve these properties, activation functions restrict the negative outputs (e.g. rectified linear unit (ReLU, (Hahnloser et al., 2000; Nair & Hinton, 2010)), leading the mean of activation outputs (i.e. activation mean) far from zero. However, negative output should proceed for zero-like activation to ensure effective and efficient training (Clevert et al., 2016; Qiu et al., 2018). Modern activation functions, such as exponential linear unit (ELU, Clevert et al. (2016)), flexible ReLU (FReLU, Qiu et al. (2018)), and sigmoid-weighted linear unit (SiLU, also known as Swish, Elfwing et al. (2018); Ramachandran et al. (2018)), seek a balance between

the properties, but could not get rid of the trade-off. Secondly, the noise-robustness varies across samples with element-level activation functions. The noise-robustness of element-level activation functions relies only on the saturation state. This implies that existing activation functions with a saturation state are expected to be robust for samples only when a sufficiently large number of elements are in the saturation state, not when there are fewer elements in the saturation state.

LayerAct functions are designed to address these issues. The trade-off problem of element-level activation functions arises because the activation input space that leads activation outputs to be in saturation state remains fixed across all samples. Unlike the element-level activation mechanism, our proposed layer-level activation mechanism assigns the saturation state based on the normalized input of the layer-dimension, similar to layer normalization (LayerNorm; Ba et al. (2016)). As a result, the activation output space of the saturation state varies between samples; the activation input space leading to saturation state is determined by the layer-dimension mean and variance. We demonstrate that the upper bound of activation fluctuation due to shift of layer input can be lower with LayerAct functions than with element-level activation functions when input is not excessively large.

Experimental analysis with the MNIST image dataset revealed the following properties of the Layer-Act functions: i) the mean activation of LayerAct functions is zero-like, and ii) the output fluctuation due to noisy input is smaller with these functions than that with element-level activation functions. We compared the performance of the residual networks (ResNets) with LayerAct functions with those with other element-level activation functions on three image classification tasks. The results on noisy CIFAR10 and CIFAR100 datasets demonstrate that LayerAct functions were superior to other element-level activation functions. Furthermore, ResNet50 with LayerAct functions also showed superior performance on both clean and noisy ImageNet datasets compared to other functions.

## 2 BACKGROUND

### 2.1 ACTIVATION SCALE

Consider a layer in a multi-layer perceptron with linear projection and an activation function. The computation of this layer, given a $r$-dimensional input vector $x = (x_1, x_2, ..., x_r)^T$, a weight matrix $W \in \mathbb{R}^{r \times d}$, and non-linear activation function $f$ is defined as follows:

$$y = W^T x, \quad a = f(y), \tag{1}$$

where $y = (y_1, y_2, ..., y_d)^T$ and $a$ are the $d$-dimensional output vectors of the linear projection and activation of a layer, respectively. The output vector $y$ of the linear projection and activation output vector $a$ (i.e. the output of the activation function) serves as the activation input (i.e. the input of the activation function) and the input of the next layer, respectively.

In some activation functions, a function bounded between one and zero characterizes the non-linearity of the activation function during forward-propagation. We define this function, denoted as $s$, and its output as the *activation scale function* and *activation scale*, respectively. The activation output during forward pass and gradient during backward pass of an element-level activation functions with activation scale function $s$ are:

$$a_i = y_i s(y_i), \quad \frac{\partial a_i}{\partial y_i} = s(y_i) + y_i \frac{\partial s(y_i)}{\partial y_i}, \tag{2}$$

where $s$ is increasing and $s(y_i) > 0$ if $y_i > 0$. For example, the activation scale functions for the $i^{th}$ element in ReLU and SiLU, are presented as follows:

$$s^{ReLU}(y_i) = \begin{cases} 1, & \text{if } y_i \geq 0 \\ 0, & \text{if } y_i < 0 \end{cases}, \quad s^{SiLU}(y_i) = \frac{1}{1 + e^{-y_i}} \tag{3}$$

where $y_i$, $sigmoid$, $s^{ReLU}$, and $s^{SiLU}$, present the $i^{th}$ element of $y$, Logistic Sigmoid function, and the non-linear scale functions of ReLU and SiLU, respectively.

Furthermore, the saturation state of such activation functions can be defined using the activation scale:

**Definition 2.1 (Saturation state of activation functions with activation scale functions)** *The saturation state of an activation function with an activation scale function $s$ is when $s(y_i) \simeq 0$, as the activation output $a_i = y_i s(y_i)$ reaches saturation.*

In conclusion, the activation scale function plays a crucial role in providing non-linearity during the forward pass, controlling the gradient during the backward pass, and determining the saturation state of the activation function.

## 2.2 TRADE-OFF BETWEEN SATURATION AND ZERO-LIKE MEAN ACTIVATION

Element-level activation functions that exhibit saturation, such as ReLU, are well recognized for their noise-robustness properties, for instance, samples with a large number of elements in the saturation state are noise-robust (Clevert et al., 2016; Qiu et al., 2018). However, the saturation in these functions does not allow negative outputs, which causes the mean of the activation outputs to be far from zero, potentially leading to inefficient training (Clevert et al., 2016).

To address this issue, recent activation functions, such as ELU, FReLU, and SiLU, saturate only the large negative outputs. These activation functions can achieve a zero-like mean activation with small negative outputs. However, a trade-off still exists because the restriction of negative outputs, designed to ensure saturation, prevents the allowance of large negative outputs, thereby restraining the mean of activation from being more zero-like. Additionally, saturation that relies solely on the input of a single element can result in a large variance in noise-robustness between samples.

## 2.3 LARGE VARIANCE OF NOISE-ROBUSTNESS ACROSS SAMPLES

To analyze the noise-robustness, we define activation fluctuation (i.e., fluctuation of activation outputs due to the shift of inputs) that can represent the layer-level noise-robustness on a sample.

**Definition 2.2 (Activation fluctuation)** *Let $\epsilon = (\epsilon_1, \epsilon_2, ..., \epsilon_d)^T$ be the noise vector. We define activation fluctuation as $\|f(y + \epsilon) - f(y)\| \leq c$, where $c$ is the upper bound of activation fluctuation.*

The lower the upper bound $c$ is, the lower the variance of noise-robustness across samples. We can define the activation fluctuation of element-level activation functions:

**Definition 2.3 (Activation fluctuation of element-level activation functions)** *Let $\epsilon_i$ be the $i^{th}$ noise, and $\hat{y}_i = y_i + \epsilon_i$. The activation fluctuation of element-level activation function $f$ is given by:*

$$\|f(\hat{y}) - f(y)\| = \sum_{i=1}^{d} |\hat{y}_i s(\hat{y}_i) - y_i s(y_i)| = \sum_{i=1}^{d} |y_i(s(\hat{y}_i) - s(y_i)) + \epsilon_i s(\hat{y}_i)|,$$

A sample will exhibit a small $\|f(\hat{y}) - f(y)\|$ if a sufficient number of its elements are in saturation state. However, element-level activation functions do not ensure that all samples have a sufficient number of elements in saturation state. More specifically, the activation fluctuation is upper-bounded when not all elements are in the saturation state, where $y_i > 0$ for all $i$:

$$\|f(\hat{y}) - f(y)\| \leq \sum_{i=1}^{d} (y_i |s(\hat{y}_i) - s(y_i)| + |\epsilon_i| \cdot s(\hat{y}_i)) \tag{4}$$

Equation 4 demonstrates that activation scale is closely related to the activation fluctuation, samples with large $\|s(\hat{y}) - s(y)\|$ and $\|s(\hat{y})\|$ are not robust to noise. Thus, a method that can reduce the upper bound of $\|s(\hat{*}) - s(*)\|$ and $\|s(\hat{*})\|$ will reduce the upper bound of activation fluctuation, resulting in a low variance of noise-robustness across samples.

## 2.4 LAYER NORMALIZATION

LayerNorm normalizes elements along the layer-dimension, as opposed to the batch-dimension in batch normalization (BatchNorm, Ioffe & Szegedy (2015)). LayerNorm normalizes the elements of a layer using the layer-dimension mean $\mu_y$ and standard deviation $\sigma_y$ defined as follows:

$$n_i^{LN} = \frac{g_i}{\sigma_y}(y_i - \mu_y) + b_i, \quad \mu_y = \frac{1}{d}\sum_{i=1}^{d} y_i, \quad \sigma_y = \sqrt{\frac{1}{d}\sum_{i=1}^{d}(y_i - \mu_y)^2} \tag{5}$$

where $n_i^{LN}$, $g_i$, and $b_i$ are the $i^{th}$ normalized output, gain, and bias of LayerNorm. With LayerNorm, the sum of activation scale $\|s\left(n^{LN}\right)\|$ will have a lower upper bound of samples, which helps to reduce the variance of noise-robustness across samples.

However, LayerNorm does not address the trade-off problem of element-level activation, and tends to exhibit poorer performance on CNNs compared to BatchNorm. LayerNorm leads the mean and variance of activation outputs to become similar across samples, as the activation output is $n_i^{LN} s\left(n_i^{LN}\right)$ when LayerNorm layers are applied before activation layers. Lubana et al. (2021) demonstrated that when a LayerNorm layer is applied before an activation layer results in similar activation output for any given network structure, thereby leading to inefficient informative propagation. Labatie et al. (2021) claimed that LayerNorm is unable to overcome the channel-wise collapse in deep networks, resulting in less efficient training compared to BatchNorm. This perspective is consistent with reports that LayerNorm shows poorer performance on CNN-based models than BatchNorm, as evidenced even in its original paper (Ba et al., 2016; Wu & He, 2018). To avoid this problem, a layer-level balancing mechanism should be employed that does not directly re-scale or re-center the activation input.

In this section, we have defined activation scale function and demonstrated its critical role in activation processes: 1) provides non-linearity during forward pass, 2) controls gradient during backward pass, and 3) is related to the noise-robustness of the model. We demonstrated that element-level activation functions may have large variance of noise-robustness across samples. LayerNorm can reduce such variance of the noise-robustness by re-scaling and re-centering the activation input, but it also causes the statistics of activation outputs to be similar across all samples.

## 3 LAYER-LEVEL ACTIVATION

In this section, we introduce and discuss a novel layer-level activation mechanism and associated functions that utilize layer-dimension normalized input for the activation scale function (see Figure 1). Our proposed method does not suffer from the trade-off issue and exhibits lower variance than element-level activation functions across samples. Importantly, it does not cause the dilution problem that statistics of activation outputs become similar.

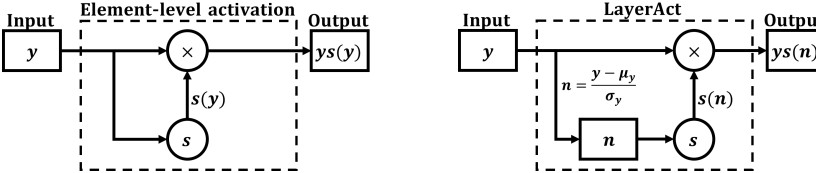

Figure 1: The mechanisms of the element-level activation (left) and proposed layer-level activation (right).

### 3.1 LAYERACT MECHANISM

The LayerAct function is defined as the product of the input $y_i$ and the activation scale $s(n_i)$ which uses the layer-normalized input $n_i$. The forward pass of a LayerAct function is given by:

$$a_i = y_i s\left(n_i\right), \quad n_i = \frac{(y_i - \mu_y)}{\sqrt{\sigma_y^2 + \alpha}} \tag{6}$$

where $\alpha > 0$ is a constant that is introduced for stability, $\mu_y$, and $\sigma_y$ are the layer-dimension mean and standard deviation, respectively. Using the chain rule, the backward pass can be described as follows:

$$\frac{\partial \mathcal{L}}{\partial \mu} = \sum_{i=1}^{d} \frac{\partial \mathcal{L}}{\partial a_i} \cdot \frac{\partial s\left(n_i\right)}{\partial n_i} \cdot \frac{-y_i}{\sqrt{\sigma^2 + \alpha}}, \qquad \frac{\partial \mathcal{L}}{\partial \sigma^2} = \sum_{i=1}^{d} \frac{\partial \mathcal{L}}{\partial a_i} \cdot \frac{\partial s\left(n_i\right)}{\partial n_i} \cdot \frac{-y_i \cdot n_i}{2\left(\sigma^2 + \alpha\right)},$$

$$\frac{\partial \mathcal{L}}{\partial y_i} = \frac{\partial \mathcal{L}}{\partial a_i} s\left(n_i\right) + \frac{\partial \mathcal{L}}{\partial a_i} \cdot \frac{\partial s\left(n_i\right)}{\partial n_i} \cdot \frac{y_i}{\sqrt{\sigma^2 + \alpha}} + \frac{1}{d} \cdot \frac{\partial \mathcal{L}}{\partial \mu} + \frac{2\left(y_i - \mu\right)}{d} \cdot \frac{\partial \mathcal{L}}{\partial \sigma^2}.$$

For stable learning and inference, it is crucial for the activation outputs to remain continuous throughout the entire output space. While element-level activation functions such as ReLU, leaky ReLU (LReLU Maas et al. (2013)), and parametric ReLU (PReLU, He et al. (2015)) do not require the activation scale to be continuous at zero (since the activation output $y_i s(y_i)$ is still continuous at zero), this is not the case for LayerAct functions, where the activation output $y_i s(n_i)$ is discontinuous if the activation scale function is not continuous. Hence, we define specific activation scale function $s$ for LayerAct mechanism:

**Definition 3.1 (Activation scale function for LayerAct functions)** *The activation scale function $s$ is an increasing Lipschitz continuous function that bounded between zero and one:*

$$s(0) = 1/2, \quad |s(a) - s(b)| \le K|a - b| \quad \forall a, b \in \mathbb{R}.$$

Any function that satisfies Definition 3.1 can be used as an activation scale function for a LayerAct function. In this paper, we suggest the Sigmoid and HardSigmoid functions as simple activation scale functions for LayerAct functions. Both the functions are Lipschitz continuous functions and bounded between $0$ and $1$. We propose the following two LayerAct functions, LA-SiLU and LA-HardSiLU, which are the layer-level transformed versions of SiLU and HardSiLU, respectively:

$$LA\text{-}SiLU(y_i) = \frac{y_i}{1 + e^{-n_i}}, \quad LA\text{-}HardSiLU(y_i) = \begin{cases} y_i, & \text{if } n_i \ge 3 \\ y_i\left(\frac{n_i}{6} + \frac{1}{2}\right), & \text{if } -3 \le n_i < 3 \\ 0, & \text{if } n_i < -3 \end{cases}.$$

LayerAct, unlike element-level activation, bypasses the trade-off between saturation and zero-like mean activation. The key distinction in saturation between the element-level and LayerAct functions is that the saturation state of element-level functions requires to be fixed at a certain point of activation output, whereas that of LayerAct functions depends on layer-dimension normalized inputs. Thus, while LayerAct still have saturation state where $s(n_i) \simeq 0$, the activation output space with a LayerAct function is not limited (e.g., consider a layer where $\mu_y \ll 0$).

Although LayerAct functions utilize layer-direction normalization for activation, the activation output $a_i$ of a LayerAct function in Equation 6 is not normalized output like that resulting from activation with LayerNorm. When a LayerNorm layer is applied right before an activation layer, the activation output would be $n_i^{LN} s(n_i^{LN})$. The re-centering and re-scaling effects of layer-direction normalization impact the activation, resulting in a loss of diversity in layer-level mean and variance statistics across samples, as discussed in Subsection 2.4. LayerAct functions only adopt the benefit of layer-direction normalization for CNNs, reducing the variance of noise-robustness across samples. For additional details and clarification, please see Appendix C.

## 3.2 Noise-robustness of LayerAct

In this subsection, we begin by establishing that the activation fluctuation of LayerAct is also related to the two terms of activation scale function, $\|s(\hat{*}) - s(*)\|$ and $\|s(\hat{*})\|$, as outlined in Subsection 2.3. Subsequently, we demonstrate that these two terms for LayerAct are bound to be lower than those of element-level activation. Here, we consider noise that is not substantial compared to activation input (i.e., $\sigma_\epsilon \ll \sigma_y$), where $\sigma_\epsilon$ represents the variance of noise $\epsilon$. To begin with, we define the activation fluctuation of LayerAct.

**Definition 3.2 (Activation fluctuation of LayerAct functions)** *The activation fluctuation of LayerAct activation function $g$, where $\hat{n}_i = (\hat{y}_i - \mu_{\hat{y}})/\sigma_{\hat{y}}$ denotes $i^{th}$ noisy normalized input, is defined as:*

$$\|g(\hat{y}) - g(y)\| = \sum_{i=1}^{d} |\hat{y}_i s(\hat{n}_i) - y_i s(n_i)| = \sum_{i=1}^{d} |y_i(s(\hat{n}_i) - s(n_i)) + \epsilon_i s(\hat{n}_i)|,$$

Given that $n$ and $\hat{n}$ represent the normalized output of $y$ and $\hat{y}$, respectively, we can define an upper bound for the activation fluctuation of LayerAct functions as follows:

$$\|g(\hat{y}) - g(y)\| \le \sum_{i=1}^{d} (|y_i| |s(\hat{n}_i) - s(n_i)| + |\epsilon_i| s(\hat{n}_i)). \tag{7}$$

Hence, the two terms of LayerAct scale function, $\|s\left(\hat{n}\right) - s\left(n\right)\|$ and $\|s\left(\hat{n}\right)\|$, are also related to the noise-robustness, similar to those of element-level activation function (see Equation 4). Considering Definition 3.1, the upper bound of $\|s\left(\hat{y}\right) - s\left(y\right)\|$ and $\|s\left(\hat{y}\right)\|$ of element-level activation and that of $\|s\left(\hat{n}\right) - s\left(n\right)\|$ and $\|s\left(\hat{n}\right)\|$ of LayerAct are given by repectively:

$$\|s\left(\hat{y}\right) - s\left(y\right)\| \leq \sum_{i=1}^{d} K\left|\epsilon_i\right|, \quad \|s\left(\hat{y}_i\right)\| \leq d, \tag{8}$$

$$\|s\left(\hat{n}\right) - s\left(n\right)\| < K\sum_{i}^{d}\left|\frac{y_i + \epsilon_i - \mu_y - \mu_\epsilon}{\sqrt{\sigma_y^2 + \alpha + \sigma_\epsilon^2}} - \frac{y_i - \mu_y}{\sqrt{\sigma_y^2 + \alpha}}\right| = \sum_{i}^{d}\frac{K\left|\epsilon_i - \mu_\epsilon\right|}{\sqrt{\sigma_y^2 + \alpha}}, \quad \|s\left(\hat{n}_i\right)\| \ll d, \tag{9}$$

where $\sqrt{\sigma_y^2 + \alpha + \sigma_\epsilon^2} \approx \sqrt{\sigma_y^2 + \alpha} > 1$ when $\sigma_y \gg \sigma_\epsilon$ and $\alpha$ is sufficiently large.

Equations 8 and 9 reveal that the activation fluctuation of LayerAct can exhibit a smaller boundary across samples compared to that of element-level activation. This implies that networks with LayerAct are likely to achieve more robust processing during the forward pass, especially when the input is not excessively large, reinforcing the importance of applying normalization methods, such as BatchNorm, in networks with LayerAct functions.

### 3.3 RELATIONSHIP BETWEEN LAYERACT AND NORMALIZATION METHODS

LayerAct does not directly re-center or re-scale its inputs, highlighting the necessity of a suitable normalization method to fully enjoy the advantages of normalization. Therefore, the choice of normalization method significantly impacts the performance of networks with LayerAct.

However, the benefit of LayerAct may be reduced when LayerNorm is placed before the activation layer, as LayerNorm's output acts as pre-normalizing for the LayerAct functions. This results in LayerAct's activation output being similar to that of the corresponding element-level activation function. Nonetheless, when an affine function is utilized in the mechanism of LayerNorm, the benefit of LayerAct can partially remain. For details, see Appendix D.

Meanwhile, LayerAct is more sensitive to the presence of similar order of elements' mean and variance (or those of channels in case of image data) in the input layer compared to element-level activation functions Thus, normalization that can prevent such similar order of statistics of elements is needed to enhance the effectiveness of LayerAct functions. Given these considerations, and recalling that the inputs to LayerAct should be excessively large to maintain noise-robustness, batch-direction normalizations such as BatchNorm or Decorrelated Batch Normalization (**?**) For details, see Appendices E) and F.

## 4 EXPERIMENTS

In this section, we present the experimental analysis and classification performance of LayerAct. First, we verify the important properties of LayerAct with the MNIST dataset. Next, we evaluate the classification performance of the LayerAct functions on three image datasets, CIFAR10, CIFAR100 (Krizhevsky, 2009), and ImageNet (Russakovsky et al., 2015) for both clean and noisy cases. We used ResNet with BatchNorm as the network architecture for our experiments (He et al., 2016). See Appendix H for details of the experimental environment, and Appendix K for more results of experiments. The tables in this section report the mean accuracy over 30 runs except the experiments on ImageNet, the best results are underlined and bolded, while the second best are bolded.

### 4.1 EXPERIMENTAL ANALYSIS ON MNIST

In this subsection, we compare the LayerAct functions with other activation functions to demonstrate that LayerAct functions embody the properties discussed in Section 3: i) zero-like mean activation and ii) noise-robustness. We trained a network with a single layer that contains 512 elements on the MNIST training dataset without any noise to observe the behavior of the LayerAct functions during training. For the details of the experimental setting, see Appendix H

### 4.1.1 Zero-like mean activation

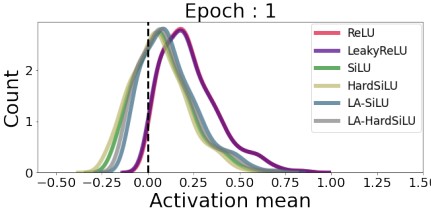 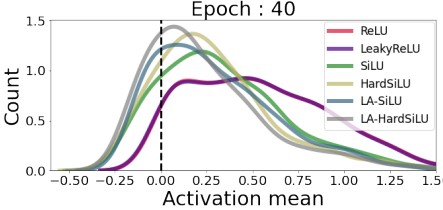

Figure 2: Distribution of the activation output means of the elements in a trained network on MNIST at 1 and 40 epochs.

Figure 2 shows the distribution of the activation output means of the single-layer network trained on the MNIST dataset at 1 and 40 epochs. The distributions did not change after 40 epochs. The LayerAct functions maintain zero-like mean activation for all epochs. Our experimental results indicate that the LayerAct functions allow similar (before epoch 20) or larger (after epoch 40) negative outputs compared to the element-level activation functions with negative outputs. Thus, LA-SiLU and LA-HardSiLU can achieve more zero-like mean activation than other activation functions.

### 4.1.2 Noise-robustness

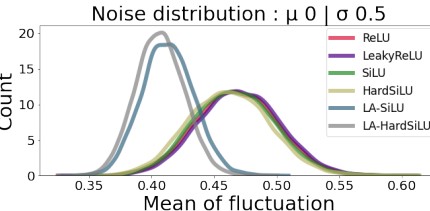 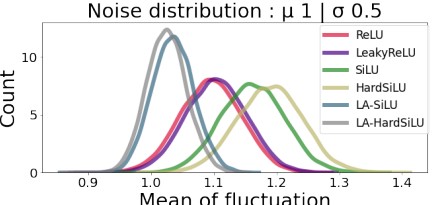

Figure 3: Distribution of activation output fluctuation due to noise with different noise distribution.

To confirm the noise-robustness of the LayerAct functions, we computed the activation fluctuation of Definition 2.3 and 3.2 using the network trained on the clean MNIST dataset. For the noisy input $\hat{y}_i$, we used two different noises with a normal distribution; one with a mean of zero and a standard deviation of 0.5, and another with a mean of one and a standard deviation of 0.5.

Figure 3 shows the distribution of the activation fluctuation with two different noise distributions. Although the fluctuation distribution of the activation input was similar (See Figure 4 in Appendix J ), LayerAct functions have a significantly smaller mean and variance of activation fluctuation among the samples than any other element-level activation function in all cases. The decrease in variance is remarkable, showing that the LayerAct functions are noise-robust for all samples. Moreover, the element-level activation functions that ensure a zero-like mean with one-sided saturation such as SiLU or HardSiLU showed slightly larger activation fluctuations than those of ReLU or LReLU when the noise had a large mean. However, the LayerAct functions maintained lower fluctuations in both cases.

## 4.2 Classification performance

We demonstrate the classification performance of the LayerAct functions on three image datasets, CIFAR10, CIFAR100, and ImageNet. We trained ResNet20, ResNet32, and ResNet44 with a basic block for CIFAR10 and CIFAR100. For ImageNet, we trained ResNet50 with the bottleneck block. In all our experiments, we utilized networks with BatchNorm. We compared the LayerAct functions with ReLU, LReLU, PReLU, Mish (Misra, 2020), SiLU and HardSiLU. We used accuracy as the performance metric. See Appendix H for the detail of experimental setting.

### 4.2.1 CLEAN CIFAR10 AND CIFAR100

The column named clean of Tables 1 and 2 presents the classification performance of ResNet20 with both LayerAct functions and element-level activation functions, benchmarked on the clean CIFAR10 and CIFAR100 dataset. In both dataset, LA-SiLU achieved the best performance among the activation functions.

In other experiments, ResNet32 and ResNet44 on clean CIFAR datasets (Table 15 in Appendix K), GELU achieved better result in two specific cases, ResNet32 on CIFAR10 and ResNet44 on CIFAR10. In the remaining combinations of networks and datasets, LA-SiLU outperformed in a significant majority of cases. Specifically, the $p$-value from T-test or Wilconxon signed-rank test was below 0.05 in 30 out of 36 experiments, indicating statistical significance. Detailed results of these statistical tests can be found in Appendix K.

Table 1: Classification performance of ResNet20 on the CIFAR10 and CIFAR10-C datasets.

| | CIFAR10 | | CIFAR10-C | | | | |
| Activation | Clean | Total | Noise | Blur | Digital | Weather | Extra |
|---|---|---|---|---|---|---|---|
| ReLU | 91.29 | 69.92 | 50.28 | 65.19 | 72.67 | 78.63 | 72.59 |
| LReLU | 91.31 | 69.82 | 49.88 | 65.26 | 72.66 | 78.41 | 72.52 |
| PReLU | 90.82 | 69.04 | 49.99 | 64.01 | 71.9 | 77.34 | 71.74 |
| SiLU | 91.45 | 70.12 | 50.23 | 65.63 | 72.54 | 78.86 | 73.03 |
| HardSiLU | 91.09 | 69.65 | 49.71 | 65.2 | 72.04 | 78.18 | 72.76 |
| Mish | 91.48 | 70.1 | 50.05 | 65.69 | 72.65 | 78.87 | 72.9 |
| GELU | **91.50** | 69.78 | 49.8 | 64.95 | 72.46 | 78.69 | 72.55 |
| ELU | 91.04 | 69.83 | 48.33 | 66.55 | 72.26 | 78.81 | 72.53 |
| LA-SiLU | **91.60** | **71.5** | **51.8** | **67.51** | **73.97** | **79.9** | **74.37** |
| LA-HardSiLU | 91.21 | **71.34** | **52.17** | 67.46 | 73.52 | 79.48 | 74.33 |

Table 2: Classification performance of ResNet20 on the CIFAR100 and CIFAR100-C datasets.

| | CIFAR100 | | CIFAR100-C | | | | |
| Activation | Clean | Total | Noise | Blur | Digital | Weather | Extra |
|---|---|---|---|---|---|---|---|
| ReLU | 65.92 | 41.81 | **21.91** | 40.41 | 43.95 | 49.02 | 42.75 |
| LReLU | 65.88 | 41.9 | 21.89 | 40.53 | 44.15 | 49.12 | 42.8 |
| PReLU | 64.00 | 39.85 | 20.71 | 38.13 | 42.35 | 46.54 | 40.71 |
| SiLU | 65.89 | 41.32 | 20.95 | 39.89 | 43.46 | 48.76 | 42.3 |
| HardSiLU | 65.19 | 41.04 | 21.31 | 39.55 | 43.19 | 48.07 | 42.11 |
| Mish | 65.85 | 41.12 | 20.48 | 39.74 | 43.39 | 48.59 | 42.08 |
| GELU | 65.84 | 41.22 | 21.02 | 39.72 | 43.31 | 48.65 | 42.2 |
| ELU | **66.24** | 41.31 | 19.17 | 40.53 | 43.63 | 49.15 | 42.32 |
| LA-SiLU | **66.39** | **42.6** | 21.48 | **41.69** | **45.03** | **50.12** | 43.47 |
| LA-HardSiLU | 66.16 | **42.85** | **22.33** | **41.76** | **45.29** | 50.06 | **43.86** |

### 4.2.2 NOISY CIFAR10 AND CIFAR100

To verify the noise-robustness of LayerAct functions, we evaluated their classification performance on the out-of-distribution benchmark datasets, CIFAR-C (Hendrycks & Dietterich, 2019). The CIFAR-C dataset includes a total of 19 distinct corruptions, each with five levels of severity, organized into five categories: noise, blur, digital, weather, and extra.

Tables 1 and 2 show the classification performance of ResNet20 on CIFAR10-C and CIFAR100-C datasets (see Appendix K for the results of ResNet32 and ResNet44). The total represents the average of accuracy across all corruptions. The networks with LayerAct functions achieved remarkable performance compared to those with element-level activation functions. In statistical significance test, networks with LA-HardSiLU outperformed those with element-level activation functions (T-test or Wilconxon signed-rank test with $p$-value$< 0.05$), except networks with ReLU and LReLU on

noisy CIFAR10 with Gaussian noise 1 and 2. This result demonstrates that LA-HardSiLU exhibits greater noise-robustness to intense noise compared to other functions. Furthermore, we experiments on ResNets without a normalization method to investigate the impact of BatchNorm on noise-robustness of LayerAct. The experiments reveals that LA-SiLU can maintain its noise-robustness when inputs are not excessively large. For the detailed results of the experiments, see Appendix E.

Table 3: Classification performance on the clean and noisy ImageNet.

| Activation | ImageNet | | | | | | |
|---|---|---|---|---|---|---|---|
| | Clean | Total | Noise | Blur | Digital | Weather | Extra |
| ReLU | 77.71 | **43.75** | **34.40** | 36.85 | **48.37** | 47.18 | 49.60 |
| LReLU | 77.83 | 43.24 | 32.87 | 36.33 | 48.00 | 47.03 | 49.38 |
| PReLU | 74.99 | 36.77 | 23.28 | 32.27 | 43.29 | 39.56 | 42.05 |
| SiLU | 77.85 | 42.31 | 29.74 | 35.94 | 46.59 | 47.36 | 48.77 |
| HardSiLU | 76.30 | 40.56 | 26.21 | 35.14 | 46.01 | 45.23 | 46.63 |
| Mish | 77.41 | 42.57 | 31.24 | 36.60 | 46.73 | 46.83 | 48.64 |
| GELU | 78.01 | 40.71 | 27.82 | 35.35 | 45.34 | 44.52 | 47.28 |
| LA-SiLU | **78.62** | **45.29** | **36.16** | **37.66** | **50.31** | **48.33** | **51.71** |
| LA-HardSiLU | **78.24** | 43.63 | 32.21 | **37.57** | 47.69 | **47.88** | **49.93** |

### 4.2.3 IMAGENET

Table 3 shows the classification performance of the LayerAct functions and the element-level activation functions for comparison with clean and noisy ImageNet datasets. We report the accuracy of 10-crop testing on validation dataset. The best results are underlined and bolded, while the second best are bolded. We used the out-of-distribution benchmark dataset, ImageNet-C (Hendrycks & Dietterich, 2019), which has the same corruptions with CIFAR-C datasets. The networks with LayerAct functions outperformed those with other activation functions on all datasets. The LayerAct functions, even LA-HardSiLU that showed worse performance on the clean CIFAR10 and CIFAR100 datasets compared to SiLU or LReLU, outperformed other activation functions on clean ImageNet.

### 4.3 FURTHER ANALYSIS AND EXPERIMENTS

The integration of LayerAct functions into a network demands careful selection of the normalization method, due to its inherent layer-direction normalizing characteristic. Our experimental results demonstrate that LayerAct is effectively compatible with BatchNorm, a normalization method prevalently employed in CNNs. Detailed analysis on the relationship between LayerAct and various normalization methods can be found in Appendices D, E, and F

To explore the viability of LayerAct, we utilized LA-SiLU in U-Net (Olaf Ronneberger, 2015) and UNet++ (Zhou et al., 2018), which are architectures designed for medical image segmentation. As shown in Table 14 in Appendix I, networks with LA-SiLU outperform those with ReLU and SiLU. These results highlight the potential of LayerAct functions in different architectures and tasks.

## 5 CONCLUSION

In this study, we introduce LayerAct that provide non-linearity with layer-direction normalizing of all elements in a layer. This unique activation mechanism achieves one-side saturation while also allowing larger negative outputs. Moreover, the activation scale with normalized input enables the LayerAct functions to reduce the mean and variance of activation fluctuation, implying that networks with LayerAct functions have potential to have lower variance of noise-robustness across samples. ResNets trained using LA-SiLU, one of the possible LayerAct functions, demonstrated similar or better performance than those for the other activation functions on the clean image datasets. Moreover, LayerAct functions outperformed the other activation functions at most of the experiments on noisy datasets. Our code and trained models are available on our GitHub repository[1].

---

[1]https://github.com/LayerAct/LayerAct

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

## A  IMPORTANT PROPERTIES OF ACTIVATION

The saturation state of an activation function contributes to robustness against shifts in activation input, as changes in saturated activation inputs minimally affect the output. However, early activation functions like Sigmoid and Tanh, which saturate on both positive and negative sides, suffer from the vanishing gradient problem. To overcome this while maintaining noise-robustness, one-sided saturation became an important property after the great success of ReLU. ReLU is expected to be noise-robust in the saturation state, (as the outputs are not affected by input shifts in the saturation state) and to have fewer vanishing gradient problems by allowing large positive outputs.

Nevertheless, functions like ReLU, which do not allow negative outputs, encounter another issue of bias shift, leading to ineffective and inefficient training. After Clevert et al. (2016) demonstrated that an activation mean closer to zero can solve such a problem, "zero-like mean activation" has become an important property of activation functions. To incorporate both properties, one-sided saturation and zero-like mean activation, activation functions such as ELU, GELU, and SiLU were designed to allow small negative outputs to push the activation mean towards zero while saturating large negative outputs for noise-robustness.

## B  DEFINITION OF ZERO-LIKE MEAN ACTIVATION

The activation output of the $i^{th}$ unit of $m^{th}$ sample ($m \in \{1, 2, ..., M\}$) is defined as $a_{i,m} = f(y_{i,m})$, where $f$, $y_{i,m}$, and $M$ are activation function, the $i^{th}$ activation input of the $m^{th}$ sample, and the number of samples, respectively. Ideally, a "zero-like activation mean" occurs when the activation mean of a single unit, $a_i$, approximates zero across the samples. Mathematically, this can be represented as:

$$\frac{1}{M} \sum_{m=1}^{M} a_{i,m} \approx 0.$$

However, approximating the activation mean to zero is challenging for the activation functions that saturate the (large) negative outputs such as ELU, SiLU or FReLU. Due to the saturation, previous studies have defined the "zero-like activation mean" property of an activation function as its ability to "push" the activation mean towards zero. In a mathematical term, this can be presented as $|\mu_{a_i}| \ll c$, where $c$ is a small positive constant (Clevert et al., 2016; Qiu et al., 2018).

## C  DIFFERENCE BETWEEN LAYERACT AND ACTIVATION WITH LAYERNORM

In this section, we compare the activation outputs between LayerAct and activation functions paired with LayerNorm. When LayerNorm is placed right before activation, the output is $a_i = n_i^{LN} s(n_i^{LN})$, where $n_i^{LN}$ is normalized output of LayerNorm. Conversely, the activation output of a LayerAct function is $a_i = y_i s(n_i)$, as defined in Equation 6 in the main article.

The critical distinction between activation with LayerNorm and LayerAct lies in the preservation of input mean and variance statistical information in the activation output. With LayerNorm, the activation function takes a layer-normalized input, resulting in activation outputs that exhibit similar statistical information across samples (as shown in the activation output equation for LayerNorm above). However, this homogenization of statistical information across samples, a characteristic of LayerNorm, is a reason why BatchNorm often outperforms LayerNorm in non-sequential models such as CNNs (Labatie et al., 2021; Lubana et al., 2021).

LayerAct, on the other hand, produces more distinguishable activation outputs between samples by preserving statistical variation between samples. This is due to the fact that only the activation scale function of LayerAct uses the layer-normalized input, not the LayerAct function itself (as shown in Equation 6 in the main article).

We would like to note that LayerAct is compatible with BatchNorm, and all the networks used in our CIFAR10, CIFAR100 and ImageNet experiments contain BatchNorm. It is worth noting that the dimension of input normalization between BatchNorm and the activation scale of LayerAct differs, which can result in different effects from BatchNorm to LayerAct. Thus, LayerAct can be effectively used with BatchNorm to enhance the performance of neural networks.

# D  LAYERACT WITH LAYERNORM

For the further investigation on the network that utilize both of LayerAct and LayerNorm, consider a network using LayerNorm without an affine function (no gain and bias) and placing the LayerNorm layer right before the activation layer. Here, the normalization output of the LayerNorm layers serves as the activation input.

Consequently, in this scenario, the normalization output of a LayerNorm layer is $n^{\hat{LN}} = \frac{y - \mu_y^2}{\sqrt{\sigma_y^2 + \alpha}}$. This leads to the output of SiLU $n^{\hat{LN}} s\left(n^{\hat{LN}}\right)$ and LA-SiLU $n^{\hat{LN}} s\left(n\right)$ to be exactly same as $n = n^{\hat{LN}}$. As such, the closer the output of the LayerNorm layer approximates the normalized input of LayerAct's activation scale function (i.e., the more gain and bias approximate to one and zero, respectively), the more the benefits of LayerAct are reduced.

However, the benefits of LayerAct functions, addressing the trade-off problem between two properties of activation and having potential to have lower variance of noise-robustness across sampels, can be partially maintain when used with LayerNorm that includes an affine function. In such scenarios, the layer-direction normalization in LayerNorm and LayerAct differs, represented as $n^{\hat{LN}} = g \cdot \frac{y - \mu_y^2}{\sqrt{\sigma_y^2 + \alpha}} + b$, and $n = \cdot \frac{y - \mu_y^2}{\sqrt{\sigma_y^2 + \alpha}}$, where $g$ and $b$ are gain and bias parameter of Layer-Norm, respectively. To explore this, we carried out experiments on ResNets with LayerNorm. We followed the same experimental setting of the experiments in our main manuscript.

Table 4: Classification performance of ResNets with LayerNorm on the CIFAR10 and CIFAR10-C datasets.

| | | **CIFAR10** | | | **CIFAR10-C** | | | |
|---|---|---|---|---|---|---|---|---|
| Model | Activation | Clean | Total | Noise | Blur | Digital | Weather | Extra |
| ResNet20 | ReLU | 88.24 | 73.77 | 62.05 | 71.26 | 76.48 | 79.69 | 76.46 |
| ResNet20 | SiLU | 87.53 | 74.3 | 63.32 | 71.74 | 77.33 | 79.77 | 76.58 |
| ResNet20 | LA-SiLU | **88.52** | **75.31** | **63.81** | **73.11** | **78.23** | **80.92** | **77.6** |
| ResNet32 | ReLU | **88.55** | 74.48 | 63.33 | 71.92 | 76.94 | 80.33 | 77.11 |
| ResNet32 | SiLU | 87.27 | 75.3 | 64.78 | 72.92 | 78.55 | 80.3 | 77.32 |
| ResNet32 | LA-SiLU | 87.85 | **76.39** | **67.35** | **74.22** | **78.80** | **80.99** | **78.32** |
| ResNet44 | ReLU | **88.58** | 75.2 | 64.32 | 72.75 | 77.87 | **80.67** | 77.67 |
| ResNet44 | SiLU | 86.65 | 75.11 | 65.67 | 72.73 | 77.84 | 79.85 | 77.1 |
| ResNet44 | LA-SiLU | 86.88 | **76.73** | **69.53** | **74.98** | **78.51** | 80.43 | **78.43** |

Table 5: Classification performance of ResNets with LayerNorm on the CIFAR100 and CIFAR100-C datasets.

| | | **CIFAR100** | | | **CIFAR100-C** | | | |
|---|---|---|---|---|---|---|---|---|
| Model | Activation | Clean | Total | Noise | Blur | Digital | Weather | Extra |
| ResNet20 | ReLU | 61.3 | 44.04 | 31.82 | 44.39 | 47.03 | 48.86 | 45.07 |
| ResNet20 | SiLU | 60.45 | **45.93** | **35.66** | **46.46** | 48.46 | 49.91 | **46.59** |
| ResNet20 | LA-SiLU | **62.3** | 45.3 | 31.63 | 45.9 | **48.72** | **50.65** | 46.17 |
| ResNet32 | ReLU | **63.21** | 45.92 | 33.23 | 46.17 | 49.26 | 50.94 | 46.84 |
| ResNet32 | SiLU | 60.04 | 47.14 | **37.37** | **48.6** | 49.57 | 50.33 | 47.41 |
| ResNet32 | LA-SiLU | 60.76 | **47.26** | 36.63 | 48.51 | **49.97** | **51.03** | **47.51** |
| ResNet44 | ReLU | **64.18** | 46.63 | 33.92 | 46.89 | **49.68** | **51.72** | **47.75** |
| ResNet44 | SiLU | 59.58 | **47.41** | **38.28** | 48.69 | 49.45 | 50.61 | 47.73 |
| ResNet44 | LA-SiLU | 60.17 | 47.31 | 38.1 | **48.82** | 49.63 | 50.26 | 47.46 |

We present the average accuracy over 10 trials in Tables 4 and 5. The experiments substantiated our concerns about the relationship between LayerNorm and LayerAct. The results showed that ResNet32 and ResNet44 with ReLU outperformed those with LA-SiLU. On the noisy CIFAR100 dataset, the performance of ResNet20 and ResNet44 with SiLU was better than those with LA-SiLU.

Nonetheless, it is noteworthy that LA-SiLU exhibited superior performance in four out of six scenarios on noisy datasets. Moreover, networks with LA-SiLU outperformed those with SiLU. This observation brings us to the conclusion that the advantages of LayerAct are diminished when paired with LayerNorm as compared to BatchNorm. However, it's noteworthy that the benefits of Layer-Act might still be partially retained when LayerNorm is utilized, particularly in scenarios where the affine function is incorporated.

## E    RELATIONSHIP BETWEEN LAYERACT AND BATCHNORM

Considering the importance of normalization methods in deep learning, exploring the relationship between LayerAct and normalization methods is needed.

Firstly, to verify whether the noise-robustness of LayerAct relies on BatchNorm, we conducted additional experiments on ResNets without a normalization method. For these experiments, we selected ReLU and SiLU as baseline activation functions for comparison with LA-SiLU. The choice of ReLU was based on its simplicity as an activation function, while SiLU was chosen due to its use of sigmoid activation scale functions, similar to LA-SiLU. We used the same experimental setting as in the experiments in our main manuscript. We present the average accuracy over 10 trials.

Table 6: Classification performance on CIFAR10 and CIFAR10-C of ResNet32 without a normaliation method. We report the average mean accuracy in the table.

| Model | Activation | CIFAR10 | CIFAR10-C | | | | | |
| | | Clean | Total | Noise | Blur | Digital | Weather | Extra |
|---|---|---|---|---|---|---|---|---|
| ResNet20 | ReLU | 88.51 | 67.43 | 53.27 | 62.12 | 70.93 | 75.91 | 71.38 |
| ResNet20 | LA-SiLU | **88.95** | **69.36** | **54.71** | **65.58** | **71.93** | **77.59** | **73.34** |
| ResNet32 | ReLU | **89.56** | 68.94 | 54.48 | 63.9 | 72.45 | 77.52 | 72.73 |
| ResNet32 | LA-SiLU | 89.12 | **70.59** | **56.29** | **66.81** | **73.11** | **78.94** | **74.22** |
| ResNet44 | ReLU | **90.03** | 69.97 | 55.75 | 65.08 | 73.38 | 78.45 | 73.62 |
| ResNet44 | LA-SiLU | 88.77 | **71.65** | **58.15** | **68.32** | **74.01** | **79.49** | **74.88** |

Table 7: Classification performance on CIFAR100 and CIFAR100-C of ResNet32 without a normaliation method. We report the average mean accuracy in the table.

| Model | Activation | CIFAR100 | CIFAR100-C | | | | | |
| | | Clean | Total | Noise | Blur | Digital | Weather | Extra |
|---|---|---|---|---|---|---|---|---|
| ResNet20 | ReLU | 88.51 | 67.43 | 53.27 | 62.12 | 70.93 | 75.91 | 71.38 |
| ResNet20 | LA-SiLU | **88.95** | **69.36** | **54.71** | **65.58** | **71.93** | **77.59** | **73.34** |
| ResNet32 | ReLU | **89.56** | 68.94 | 54.48 | 63.9 | 72.45 | 77.52 | 72.73 |
| ResNet32 | LA-SiLU | 89.12 | **70.59** | **56.29** | **66.81** | **73.11** | **78.94** | **74.22** |
| ResNet44 | ReLU | **90.03** | 69.97 | 55.75 | 65.08 | 73.38 | 78.45 | 73.62 |
| ResNet44 | LA-SiLU | 88.77 | **71.65** | **58.15** | **68.32** | **74.01** | **79.49** | **74.88** |

Table 6 and 7 demonstrates the performance of ResNets without any normalization on both clean and noisy CIFAR10 and CIFAR100. We used a learning rate of 0.01. We report the average classification accuracy among 30 trials. ResNet32 and ResNet44 with SiLU exploded during training in some trials. Thus, we present the experimental results of the networks with ReLU and LA-SiLU.

The networks with LA-SiLU outperformed the networks with ReLU on noisy datasets. This result and the one from Appendix D imply that the noise-robustness of LA-SiLU is not depend on the normalization methods, if the activation input is not excessively large.

However, the performance of ResNet32 and ResNet44 with ReLU outperformed those with LA-SiLU on clean datasets. This is because networks with LA-SiLU sufer the overfitting problem, showing 95.76% and 96.47% accuracy on CIFAR10 and CIFAR100 train dataset, repsectively, which are higher than those with ReLU, 92.23% and 94.52%. This reveals that deeper networks with LA-SiLU needs the re-scaling and re-centering operation of normalization methods, such as BatchNorm, to prevent overfitting.

The efficacy of BatchNorm extends beyond its re-scaling and re-centering operations and is also attributed to its directional approach. When considering an image dataset $X \in \mathbb{R}^{n \times c \times h \times w}$, batch-direction normalization introduces dynamic variablility to the order of mean and variance of channels across sampels. This means that the order of mean and variance in channles can differ between samples, contributing to the unique advantages of BatchNorm. This mechanism of BatchNorm is very helpful to LayerAct, which uses layer-direction normalizing for activation and is highly sensitive to the sequence in which mean and variance are calculated across channels.

In conclusion, selecting normalization method for networks with LayerAct functions should be in precise way, considering the operation of both LayerAct and normalization method. While Layer-Norm can reduce the benefit of LayerAct as discussed in Appendix D, while BatchNorm have good properties, BatchNorm would be the one for the networks with LayerAct functions on image dataset.

## F LAYERACT WITH OTHER NORMALIZATION METHODS.

To investigate the relationship between LayerAct and normalizations, we conducted experiments on ResNets with Switchable Normalization (SwitchNorm, Luo et al. (2019)), Instance enhancement batch normalization (IEBN, Liang et al. (2020)), and Decorrelated Batch Normalization (DBN, Huang et al. (2018)). We used same experiment setting with those of our main manuscript. We report the average accuracy over 10 runs for the experiments with SwitchNorm and IEBN, and average accuracy over 5 runs for those with DBN.

Analysis of the experimental results revealed that LayerAct functions are not effectively compatible with normalizations that can cause a large variance in the channel means, such as SwitchNorm and IEBN. This incompatibility arises because LayerAct is more sensitive to such large variance between channel means, which resulting channels with smaller means to be more likely to become inactivated compared to those with larger means.

Table 8: Classification performance on CIFAR10 and CIFAR10-C of ResNet32 with SwitchNorm. We report the average mean accuracy in the table.

| | | CIFAR10 | | CIFAR10-C | | | | |
| Model | Activation | Clean | Total | Noise | Blur | Digital | Weather | Extra |
|---|---|---|---|---|---|---|---|---|
| ResNet20 | ReLU | 89.65 | 72.4 | 56.07 | 71.35 | 75.24 | 80.43 | 74.81 |
| ResNet20 | SiLU | **90.6** | **74.17** | **57.7** | **72.94** | **77.57** | **82.21** | **76.33** |
| ResNet20 | LA-SiLU | 89.56 | 72.43 | 55.9 | 71.05 | 75.47 | 80.87 | 74.73 |
| ResNet32 | ReLU | 90.7 | 73.9 | 58.24 | 72.84 | 76.3 | 81.79 | 76.42 |
| ResNet32 | SiLU | **90.79** | **74.65** | **58.78** | **73.39** | **77.64** | **82.57** | **76.91** |
| ResNet32 | LA-SiLU | 89.94 | 72.72 | 56.15 | 71.07 | 75.52 | 81.48 | 75.23 |
| ResNet44 | ReLU | **91.4** | **74.65** | **58.1** | **74.01** | **77.18** | **82.81** | **76.99** |
| ResNet44 | SiLU | 74.48 | 61.7 | 49.5 | 60.58 | 64.11 | 67.87 | 63.41 |
| ResNet44 | LA-SiLU | 89.36 | 72.02 | 54.85 | 70.59 | 75.07 | 81.05 | 74.24 |

Table 9: Classification performance on CIFAR100 and CIFAR100-C of ResNet32 with Switch-Norm. We report the average mean accuracy in the table.

| | | CIFAR100 | | CIFAR100-C | | | | |
| Model | Activation | Clean | Total | Noise | Blur | Digital | Weather | Extra |
|---|---|---|---|---|---|---|---|---|
| ResNet20 | ReLU | 57.36 | 37.01 | 20.61 | 38.43 | 39.75 | 43.56 | 38.59 |
| ResNet20 | SiLU | **64.55** | **42.96** | **25.13** | **43.91** | **46.35** | **50.42** | **44.55** |
| ResNet20 | LA-SiLU | 63.88 | 41.76 | 23.23 | 42.83 | 45.28 | 49.43 | 43.4 |
| ResNet32 | ReLU | 60.19 | 38.81 | 21.89 | 40.02 | 41.3 | 46.0 | 40.61 |
| ResNet32 | SiLU | **64.35** | **42.45** | **25.00** | 43.08 | **45.86** | 49.92 | **44.00** |
| ResNet32 | LA-SiLU | 64.05 | 42.27 | 23.72 | **43.46** | 45.5 | **50.16** | 43.89 |
| ResNet44 | ReLU | 61.64 | 39.93 | 22.64 | 41.03 | 42.47 | 47.54 | 41.67 |
| ResNet44 | SiLU | 44.8 | 29.57 | 18.11 | 29.96 | 31.98 | 34.21 | 30.73 |
| ResNet44 | LA-SiLU | **62.99** | **41.91** | **22.86** | **43.06** | **46.0**2 | **49.79** | **43.06** |

Table 8 and 9 exhibit that the combination of SwitchNorm and LayerAct is not effective. It is because LayerAct is more sensitive to the presence of similar channel characteristics across samples compared to element-level activation functions. If two samples display similar orders in the mean and variance of their channels, LayerAct, which employs layer-direction normalization for activation, tends to yield similar activation outputs.

BatchNorm distinctively ensure different channel characteristics between samples. While LayerNorm does not inherently differentiate channels across samples, it does not actively homogenize them either, thus preserving the natural order of mean and variance among channels. On the other hand, InstanceNorm actively normalizes the mean and variance of channels to be more uniform. SwitchNorm utilize the weighted average of three normalization method. Consequently, due to the integrated normalization approach of SwitchNorm, which combines three methods, we expect that the effect of InstanceNorm tends to homogenize channel characteristics, rendering LayerAct functions susceptible to inefficiency.

Table 10: Classification performance on CIFAR10 and CIFAR10-C of ResNet32 with IEBN. We report the average mean accuracy in the table.

| | | CIFAR100 | | CIFAR100-C | | | |
|---|---|---|---|---|---|---|---|
| Model | Activation | Clean | Total | Noise | Blur | Digital | Weather | Extra |
| ResNet20 | ReLU | 91.5 | 70.12 | 52.49 | 66.73 | 73.44 | 79.64 | 73.89 |
| ResNet20 | SiLU | 91.44 | 68.98 | 49.39 | 66.26 | 72.29 | 79.0 | 73.06 |
| ResNet20 | LA-SiLU | **91.63** | **70.64** | **52.7** | **67.3** | **74.2** | **79.87** | **74.65** |
| ResNet32 | ReLU | textbf92.54 | 71.67 | 54.06 | **68.82** | 74.82 | 81.12 | 75.11 |
| ResNet32 | SiLU | 91.85 | 71.07 | 53.39 | 68.26 | 73.9 | 80.5 | 74.9 |
| ResNet32 | LA-SiLU | 92.36 | **71.77** | **54.19** | 68.19 | **75.32** | **81.15** | **75.6** |
| ResNet44 | ReLU | **92.78** | 72.18 | 55.2 | 68.83 | 75.61 | 81.59 | 75.43 |
| ResNet44 | SiLU | 92.08 | 71.23 | 52.79 | 68.54 | 74.27 | 80.95 | 74.99 |
| ResNet44 | LA-SiLU | 92.5 | **73.01** | **56.97** | **69.18** | **76.03** | **82.34** | **76.5** |

Table 11: Classification performance on CIFAR100 and CIFAR100-C of ResNet32 with IEBN. We report the average mean accuracy in the table.

| | | CIFAR100 | | CIFAR100-C | | | |
|---|---|---|---|---|---|---|---|
| Model | Activation | Clean | Total | Noise | Blur | Digital | Weather | Extra |
| ResNet20 | ReLU | 66.62 | **41.97** | **22.81** | **42.51** | 45.07 | 50.3 | 44.37 |
| ResNet20 | SiLU | 66.19 | 40.88 | 21.14 | 41.08 | 44.29 | 49.68 | 43.28 |
| ResNet20 | LA-SiLU | **66.73** | 41.59 | 21.31 | 41.98 | **45.26** | **50.53** | **43.82** |
| ResNet32 | ReLU | **68.17** | 43.49 | 23.81 | **43.89** | 46.76 | 52.3 | **45.78** |
| ResNet32 | SiLU | 67.43 | 42.3 | 23.15 | 42.38 | 45.33 | 51.14 | 44.68 |
| ResNet32 | LA-SiLU | 67.97 | **43.56** | **24.49** | 43.39 | **46.94** | **52.47** | 45.76 |
| ResNet44 | ReLU | **69.47** | 44.73 | 25.74 | 44.77 | 47.77 | 53.51 | 47.12 |
| ResNet44 | SiLU | 68.18 | 43.47 | 24.76 | 43.37 | 46.46 | 52.22 | 45.89 |
| ResNet44 | LA-SiLU | 68.41 | **45.29** | **26.89** | **45.15** | **48.67** | **54.03** | **47.13** |

Table 10 and 11 demonstrate the preformacne of networks with IEBN. With the exception of ResNet20 on CIFAR100, networks with LA-SiLU demonstrated enhanced performance on noisy datasets when compared to their counterparts utilizing ReLU and SiLU. Conversely, ReLU outperformed LA-SiLU in ResNet32 and ResNet44 models. Nonetheless, it is important to highlight that LA-SiLU consistently surpassed SiLU, which utilizes the same activation scale function, across all tested scenarios.

These results imply that the mechanism of LayerAct holds promise for enhancing efficiency. It is also important to consider careful consideration and integration of network complexity, particularly due to the interplay between normalization and activation functions, are essential, given that the scale function of ReLU is considerably simpler compared to sigmoid, the scale function of LA-SiLU and SiLU.

Table 12 and 13 demonstrate the preformacne of networks with DBN. Except for ResNet20 on CIFAR100, networks with LA-SiLU showed similar or improved performance on both clean and noisy

Table 12: Classification performance on CIFAR10 and CIFAR10-C of ResNet32 with DBN. We report the average mean accuracy in the table.

| Model | Activation | CIFAR100 Clean | Total | Noise | CIFAR100-C Blur | Digital | Weather | Extra |
|---|---|---|---|---|---|---|---|---|
| ResNet20 | ReLU | 89.96 | 64.9 | 35.84 | 64.19 | 71.27 | 77.93 | 67.99 |
| ResNet20 | SiLU | 89.69 | 65.02 | 39.63 | 63.08 | 70.24 | 76.98 | 68.83 |
| ResNet20 | LA-SiLU | **91.33** | **68.49** | **43.02** | **67.59** | **73.41** | **79.73** | **72.33** |
| ResNet32 | ReLU | 91.99 | 68.18 | 37.10 | 68.58 | **75.0** | 81.30 | 71.15 |
| ResNet32 | SiLU | **92.27** | 68.02 | 40.16 | 66.64 | 73.86 | 80.91 | 71.57 |
| ResNet32 | LA-SiLU | 92.26 | **70.33** | **44.43** | **69.65** | **75.0** | **82.12** | **73.99** |
| ResNet44 | ReLU | 92.30 | 69.07 | 38.99 | 69.20 | **75.73** | 81.82 | 72.07 |
| ResNet44 | SiLU | 92.13 | 68.89 | 41.72 | 68.09 | 74.14 | 81.17 | 72.52 |
| ResNet44 | LA-SiLU | **92.42** | **70.94** | **46.53** | **69.58** | 75.17 | **82.61** | **74.69** |

Table 13: Classification performance on CIFAR100 and CIFAR100-C of ResNet32 with DBN. We report the average mean accuracy in the table.

| Model | Activation | CIFAR100 Clean | Total | Noise | CIFAR100-C Blur | Digital | Weather | Extra |
|---|---|---|---|---|---|---|---|---|
| ResNet20 | ReLU | **59.71** | **34.19** | 13.48 | **34.26** | **38.28** | **43.67** | **36.07** |
| ResNet20 | SiLU | 57.82 | 32.13 | 13.41 | 31.57 | 35.78 | 40.99 | 34.22 |
| ResNet20 | LA-SiLU | 58.19 | 32.24 | **14.32** | 30.95 | 35.11 | 41.4 | 34.96 |
| ResNet32 | ReLU | 54.06 | 30.29 | 14.16 | 28.67 | 34.09 | 38.34 | 32.14 |
| ResNet32 | SiLU | 62.0 | 35.26 | 14.99 | 34.41 | 39.35 | 45.01 | 37.46 |
| ResNet32 | LA-SiLU | **65.02** | **38.75** | **19.18** | **37.84** | **41.81** | **48.66** | **41.36** |
| ResNet44 | ReLU | 60.32 | 34.64 | 14.96 | 33.68 | 39.03 | 44.1 | 36.51 |
| ResNet44 | SiLU | 64.67 | 37.58 | 15.81 | 37.63 | 42.01 | 47.2 | 39.83 |
| ResNet44 | LA-SiLU | **67.59** | **42.44** | **21.03** | **42.54** | **46.52** | **52.15** | **44.6** |

datasets compared to those employing ReLU and LA-SiLU. The results of these experiments highlight the potential applicability of LayerAct functions in conjunction with advanced batch-direction normalization methods.

## G   ACTIVATION OUTPUT OF LAYERACT FUNCTIONS

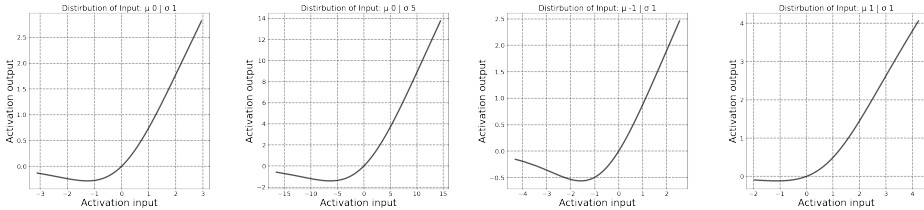

Figure 4: LA-SiLU with different mean and variance values in the input. The distribution of the activation input is: i) $\mu_y = 0$, $\sigma_y = 1$, ii) $\mu_y = 0$, $\sigma_y = 5$, iii) $\mu_y = -5$, $\sigma_y = 1$, and iv) $\mu_y = 5$, $\sigma_y = 1$ from the left to right.

In this section, we present and discuss an illustration of LayerAct functions. Unlike other activation functions, the mean and variance of the input affect the shape of the activation output in the LayerAct functions (as outlined in Equation 6 in the main article). For better demonstration of this characteristic, we present the outputs of the LayerAct functions for four distinct cases. Each case uses an input that follows a different normal distribution.

Figures 4 and 5 plot the activation outputs of LA-SiLU and LA-HardSiLU, respectively. These figures demonstrate how the shape of activation output is different depending on the shape, mean

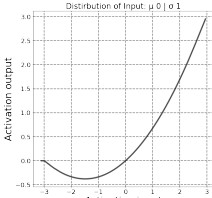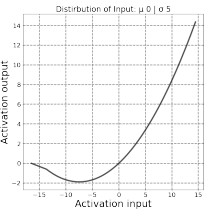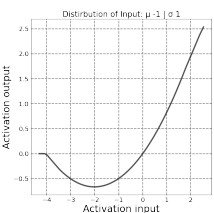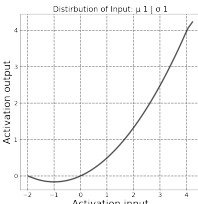

Figure 5: LA-HardSiLU with different mean and variance values in the input. The distribution of the activation input is: i) $\mu_y = 0$, $\sigma_y = 1$, ii) $\mu_y = 0$, $\sigma_y = 5$, iii) $\mu_y = -5$, $\sigma_y = 1$, and iv) $\mu_y = 5$, $\sigma_y = 1$ from the left to right.

and variance in this case, of the activation input. The figures also show that LayerAct functions can produce negative outputs depending on the mean and variance of the inputs. In some cases, no output exists in the saturation state (see the second figure in Figure 5). It is notable that the LayerAct functions achieved noise-robustness without a large number of elements in the saturation state.

## H  EXPERIMENTAL REPRODUCTION

We implemented LayerAct functions and networks for experiment with PyTorch (Paszke et al., 2019). All networks used in our experiments were trained on NVIDIA A100. We used multiple devices to train the networks on ImageNet, and a single device for the other experiments. The versions of Python and the packages were i) Python 3.9.12, ii) numpy 1.19.5 iii) PyTorch 1.11.0, and iv) torchvision 0.12.0. We used cross entropy loss functions for all the experiments. The random seeds of the experiments were $11 \times i$ where $i \in \{1, 2, ..., 30\}$ on CIFAR10 and CIFAR100 and 11 and 22 on ImageNet.

To train networks on MNIST for experimental analysis, we applied batch gradient descent for 80 epochs with the weight decay and momentum fixed to 0.0001 and 0.9, respectively. The learning rate started from 0.01, and was multiplied 0.1 at epochs 40 and 60 as scheduled.

We used ResNet (He et al., 2016) with BatchNorm right before activation for experiments on CI-FAR10, CIFAR100 and ImageNet. We initialized the weights following the methods proposed by He et al. (He et al., 2015). For all experiments, the weight decay, momentum, and initial learning rate were 0.0001, 0.9 and 0.1, respectively.

For CIFAR10 and CIFAR100, we trained ResNet20, ResNet32, and ResNet44 with a basic block using the stochastic gradient descent with a batch size of 128 for about 64000 iterations. We randomly selected 10% of the training dataset as the validation set. The learning rate was scheduled to decrease by the factor of 10 at 32000 and 48000 iterations. For the data augmentation of CIFAR10 and CIFAR100, we followed Lee et al. (2015). We rescaled the data between 0 and 1, padded 4 pixels on each side, and randomly sampled a $32 \times 32$ crop from the padded image or its horizontal flip. The data was normalized after augmentation. For testing, we did not apply data augmentation, only normalized the data. The hyper-parameter $\alpha$ of LayerAct functions for the experiments was set to 0.00001.

For the experiment with ImageNet, we trained ResNet50 with the bottleneck block using stochastic gradient descent, and the batch size was 256 for about 600000 iterations. The learning rate was scheduled to decrease by a factor of 10 at 180000, 360000, and 540000 iterations. For the data augmentation on ImageNet, we rescaled the data between 0 and 1, resized it to $224 \times 244$, and randomly sampled a $224 \times 224$ crop from an image or its horizontal flip (Krizhevsky et al., 2017). We normalized the data after data augmentation. For testing, we resized the data to be $256 \times 256$ and applied 10-crop. Afterward, the data was normalized. To ensure stable learning, we set the hyper-parameter $\alpha$ of LayerAct functions to 0.1 which is larger than those for CIFAR10 and CIFAR100.

The noisy datasets were generated by adding noise to the data after it was rescaled between 0 and 1. Following this, the same data augmentation applied to the clean dataset was also used on the noisy dataset.

The supplementary material of this paper and the trained networks are available in our anonymous GitHub repository[2].

## I  U-NET AND UNET++ WITH LA-SILU ON A MEDICAL IMAGE SEGMENTATION TASK.

In this section, we present the experimental results from U-net (Olaf Ronneberger, 2015) and Unet++ (Zhou et al., 2018) for segmentation task on a nuclei image dataset from Data Science Bowl 2018 (Goodman et al., 2018). Detailed experimental setting is as follows: i) Adam optimizer with $3e^{-4}$ learning rate, and $1e^{-4}$ weight decay, ii) training 100 epoches with cosine annealing schedular, and iii) BCE-Dice Loss as the loss function. We report the average IoU (Intersection over Union; %) over 10 trials with different weight initialization.

Table 14: Segmentation performance on U-net and Unet++ of ResNet32. We report the average mean accuracy in the table.

| Activation | U-net | Unet++ w/o DSV | Unet++ with DSV |
|---|---|---|---|
| ReLU | 84.71 | 84.94 | 84.92 |
| SiLU | 84.87 | 85.15 | 85.01 |
| LA-SiLU | **85.13** | **85.27** | **85.05** |

DSV and w/o indicate deep supervision (Lee et al., 2015) and without, respectively. The experimental results demonstrate that networks with LA-SiLU outperform those with ReLU and SiLU in every case. This reveals the practical potential of LayerAct functions across CNN-based architectures and image segmentation tasks. It is also important to highlight that neither UNet nor UNet++ utilizes a normalization method. Considering the experimental results of networks without a normalization in Appendix E , where ResNet20 with LA-SiLU demonstrated better performance compared to that with ReLU and SiLU, this suggests that LayerAct can exhibit robust performance in shallow networks without normalization.

## J  ADDITIONAL FIGURES

In this section, we present additional tables and figures extracted from the experiments.

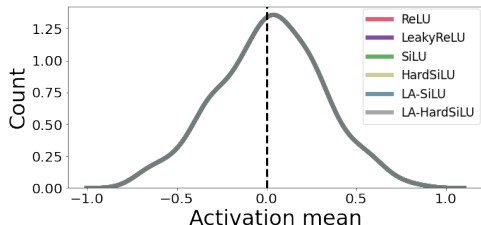 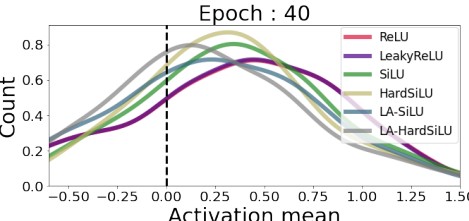

Figure 6: Distribution of the activation **input** means of the elements in a trained network on MNIST at $1^{st}$ and $40^{th}$ epochs.

Figure 6 presents the distribution of the mean activation input. As observed in the mean of activation input at epoch 40 (right), LayerAct functions promote the training of parameter $W$ such that the output of the linear projection $y = W^T x$, which is also activation input, gets closer to zero compared to other functions. This helps the activation output to exhibit a 'zero-like' behaviour.

LayerAct functions exhibit a significantly lower mean and variance of activation fluctuation among the samples compared to any other element-level activation function (see Figure 3 in the main article). Figure 7 demonstrates that the distribution of mean fluctuation in activation input appears

---

[2]https://github.com/LayerAct/LayerAct

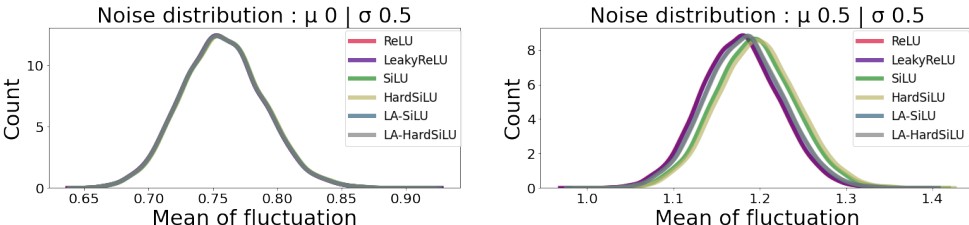

Figure 7: Distribution of activation **input** fluctuation due to noise with different noise distribution.

similar across all functions. This observation confirms that the lower mean and variance of activation output fluctuation of LayerAct functions is not due to a smaller fluctuation in activation input, but is a result of the inherent mechanism of LayerAct.

## K ADDITIONAL TABLES

Table 15 shows the classification performance of networks on clean CIFAR10 and CIFAR100 datasets.

Tables 16, 17, 18, and 19 demonstrate the classification performance of ResNet32 and ResNet44 with activation functions on CIFAR10-C and CIFAR100-C datasets. We do not report the experiments of ResNet44 with PReLU on CIFAR10 as a network exploded during training. The performance of networks with LayerAct functions were better or similar compared to other activation functions.

Table 15: Classification performance on the clean CIFAR10 and CIFAR100

| Activation | CIFAR10 | | | CIFAR100 | | |
| --- | --- | --- | --- | --- | --- | --- |
| | ResNet20 | ResNet32 | ResNet44 | ResNet20 | ResNet32 | ResNet44 |
| ReLU | 91.29 | 92.03 | 92.03 | 65.92 | 67.04 | 68.02 |
| LReLU | 91.31 | 92.03 | 92.03 | 65.88 | **67.37** | 67.96 |
| PReLU | 90.82 | 92.03 | - | 64.00 | 66.35 | 67.68 |
| SiLU | 91.45 | 92.17 | 92.18 | 65.89 | 67.22 | 67.71 |
| HardSiLU | 91.09 | 91.77 | 91.42 | 65.19 | 66.49 | 66.38 |
| Mish | 91.48 | **92.21** | 92.30 | 65.85 | 67.18 | 68.06 |
| GELU | **91.50** | 92.25 | **92.22** | 65.84 | 67.30 | **68.19** |
| ELU | 91.04 | 91.61 | 91.68 | **66.24** | 67.01 | 67.55 |
| LA-SiLU | **91.60** | 92.20 | **92.36** | **66.39** | **67.74** | **68.07** |
| LA-HardSiLU | 91.21 | 91.68 | 91.36 | 66.16 | 66.63 | 65.51 |

Table 16: Classification performance on CIFAR10 and CIFAR10-C of ResNet32. We report the average mean accuracy in the table.

| Activation | Clean | Total | Noise | Blur | Digital | Weather | Extra |
| --- | --- | --- | --- | --- | --- | --- | --- |
| ReLU | 92.03 | 72.00 | 53.07 | 67.62 | 74.75 | 80.44 | 74.41 |
| LReLU | 92.03 | 72.01 | 52.66 | **67.86** | **74.77** | 80.42 | 74.5 |
| PReLU | 92.03 | 71.7 | 52.82 | 67.06 | 74.72 | 79.90 | 74.17 |
| SiLU | 92.17 | 71.7 | 52.37 | 67.21 | 74.08 | 80.51 | 74.38 |
| HardSiLU | 91.77 | 71.32 | 52.75 | 66.75 | 73.39 | 79.66 | 74.28 |
| Mish | **92.21** | 71.96 | 53.09 | 67.42 | 74.3 | 80.57 | 74.64 |
| GELU | 92.25 | 71.64 | 52.44 | 67.18 | 74.11 | 80.26 | 74.26 |
| LA-SiLU | 92.20 | **72.8** | 54.02 | **68.42** | **75.13** | **81.36** | **75.51** |
| LA-HardSiLU | 91.68 | **72.6** | **55.36** | 67.71 | 74.70 | **80.53** | **75.62** |

Table 17: Classification performance on CIFAR10 and CIFAR10-C of ResNet44. We report the average mean accuracy in the table.

| Activation | Clean | Total | Noise | Blur | Digital | Weather | Extra |
| --- | --- | --- | --- | --- | --- | --- | --- |
| ReLU | 92.03 | **73.71** | **56.39** | **70.03** | **76.05** | 81.27 | 75.92 |
| LReLU | 92.03 | **73.69** | 56.03 | **70.10** | **76.09** | **81.43** | 75.81 |
| SiLU | 92.18 | 72.45 | 53.12 | 68.64 | 74.80 | 80.88 | 75.06 |
| HardSiLU | 91.42 | 72.63 | 55.51 | 68.72 | 74.73 | 79.86 | 75.34 |
| Mish | 92.30 | 72.79 | 53.74 | 68.86 | 75.22 | 81.18 | 75.3 |
| GELU | **92.22** | 72.82 | 54.65 | 68.69 | 75.26 | 80.85 | 75.24 |
| LA-SiLU | **92.36** | 73.5 | 55.29 | 69.14 | 75.73 | **81.91** | **76.19** |
| LA-HardSiLU | 91.36 | 73.33 | **57.45** | 68.48 | 75.30 | 80.64 | **76.30** |

Table 18: Classification performance on CIFAR100 and CIFAR100-C of ResNet32. We report the average mean accuracy in the table.

| Activation | Clean | Total | Noise | Blur | Digital | Weather | Extra |
|---|---|---|---|---|---|---|---|
| ReLU | 67.04 | 43.51 | 24.10 | 41.99 | 45.81 | 50.58 | 44.32 |
| LReLU | **67.37** | 43.58 | 23.8 | 42.12 | 45.94 | 50.73 | 44.42 |
| PReLU | 66.35 | 42.44 | 23.72 | 40.31 | 44.81 | 49.42 | 43.27 |
| SiLU | 67.22 | 42.94 | 23.06 | 41.27 | 45.01 | 50.31 | 44.02 |
| HardSiLU | 66.49 | 42.67 | 23.64 | 40.87 | 44.87 | 49.54 | 43.71 |
| Mish | 67.18 | 42.95 | 22.69 | 41.53 | 45.05 | 50.37 | 43.97 |
| GELU | 67.30 | 43.00 | 23.15 | 41.45 | 45.21 | 50.21 | 43.94 |
| LA-SiLU | **_67.74_** | **44.6** | **24.16** | **43.58** | **46.74** | **_52.11_** | **45.51** |
| LA-HardSiLU | 66.63 | **_44.86_** | **_25.98_** | **_43.83_** | **_47.02_** | **51.50** | **_45.81_** |

Table 19: Classification performance on CIFAR100 and CIFAR100-C of ResNet44. We report the average mean accuracy in the table.

| Activation | Clean | Total | Noise | Blur | Digital | Weather | Extra |
|---|---|---|---|---|---|---|---|
| ReLU | 68.02 | 44.77 | 25.45 | 43.05 | 47.33 | 51.94 | 45.44 |
| LReLU | 67.96 | 44.8 | 25.57 | 43.26 | 47.17 | 51.91 | 45.5 |
| PReLU | 67.68 | 44.31 | 25.78 | 42.19 | 46.7 | 51.44 | 44.97 |
| SiLU | 67.71 | 44.04 | 24.52 | 42.61 | 46.16 | 51.08 | 45.01 |
| HardSiLU | 66.38 | 44.11 | 26.22 | 42.77 | 46.29 | 50.32 | 44.9 |
| Mish | 68.06 | 44.14 | 24.18 | 42.69 | 46.36 | 51.37 | 45.11 |
| GELU | **_68.19_** | 43.93 | 24.39 | 42.29 | 46.15 | 51.02 | 44.85 |
| LA-SiLU | **68.07** | **46.12** | **26.68** | **44.94** | **48.32** | **_53.44_** | **46.89** |
| LA-HardSiLU | 65.51 | **_46.83_** | **_30.85_** | **_45.83_** | **_48.93_** | **52.5** | **_47.35_** |

Tables 20 and 21 present the results of a statistical significance test between the accuracy of networks with element-level activation functions and those with LA-SiLU functions on clean CIFAR10 and CIFAR100. Tables 22 and 23 present the corresponding results of networks with LA-SiLU and LA-HardSiLU on CIFAR10-C and CIFAR100-C. RN20, RN32, and RN44 denotes ResNet20, ResNet32, and ResNet44. We do not report the experiments of ResNet44 with PReLU on CIFAR10 and CIFAR10-C as a network exploded during training. When the accuracies of both functions were normally distributed, we performed a T-test. In cases where at least one of them are not, we performed a Wilconxon signed-rank test otherwise. The notation '¿0.05' indicates that the $p$-value from either a T-test or a Wilcoxon signed-rank test is larger than the standard significance level of 0.05 (i.e. $p$-value $> 0.05$).

Table 20: Statistical significance test of LA-SiLU on CIFAR10 dataset.

| | ReLU | LReLU | PReLU | SiLU | HardSiLU | Mish | GELU | ELU |
|---|---|---|---|---|---|---|---|---|
| | | | | **LA-SiLU** | | | | |
| RN20 | $< 1e^{-3}$ | $< 1e^{-3}$ | $< 1e^{-3}$ | 0.002 | $< 1e^{-3}$ | 0.011 | 0.016 | $< 1e^{-3}$ |
| RN32 | 0.02 | 0.011 | 0.005 | 0.331 | $< 1e^{-3}$ | 0.422 | 0.125 | $< 1e^{-3}$ |
| RN44 | 0.014 | 0.001 | - | 0.007 | $< 1e^{-3}$ | 0.479 | 0.094 | $< 1e^{-3}$ |

Table 21: Statistical significance test of LA-SiLU on CIFAR100 dataset.

| | ReLU | LReLU | PReLU | SiLU | HardSiLU | Mish | GELU | ELU |
|---|---|---|---|---|---|---|---|---|
| | | | | **LA-SiLU** | | | | |
| RN20 | $< 1e^{-3}$ | $< 1e^{-3}$ | $< 1e^{-3}$ | $< 1e^{-3}$ | $< 1e^{-3}$ | $< 1e^{-3}$ | $< 1e^{-3}$ | 0.065 |
| RN32 | $< 1e^{-3}$ | $< 1e^{-3}$ | $< 1e^{-3}$ | $< 1e^{-3}$ | $< 1e^{-3}$ | $< 1e^{-3}$ | $< 1e^{-3}$ | $< 1e^{-3}$ |
| RN44 | 0.357 | 0.244 | 0.008 | 0.006 | $< 1e^{-3}$ | 0.476 | 0.207 | $< 1e^{-3}$ |

Table 22: Statistical significance test of LayerAct functions on CIFAR10-C dataset.

| | ReLU | LReLU | PReLU | SiLU | HardSiLU | Mish | GELU | ELU |
|---|---|---|---|---|---|---|---|---|
| | | | | **LA-SiLU** | | | | |
| RN20 | $< 1e^{-3}$ | $< 1e^{-3}$ | $< 1e^{-3}$ | $< 1e^{-3}$ | $< 1e^{-3}$ | $< 1e^{-3}$ | $< 1e^{-3}$ | $< 1e^{-3}$ |
| RN32 | $< 1e^{-3}$ | $< 1e^{-3}$ | $< 1e^{-3}$ | $< 1e^{-3}$ | $< 1e^{-3}$ | $< 1e^{-3}$ | $< 1e^{-3}$ | $< 1e^{-3}$ |
| RN44 | 0.399 | 0.233 | - | $< 1e^{-3}$ | 0.001 | 0.002 | 0.003 | $< 1e^{-3}$ |
| | | | | **LA-HardSiLU** | | | | |
| RN20 | $< 1e^{-3}$ | $< 1e^{-3}$ | $< 1e^{-3}$ | $< 1e^{-3}$ | $< 1e^{-3}$ | $< 1e^{-3}$ | $< 1e^{-3}$ | $< 1e^{-3}$ |
| RN32 | 0.007 | 0.014 | $< 1e^{-3}$ | $< 1e^{-3}$ | $< 1e^{-3}$ | $< 1e^{-3}$ | $< 1e^{-3}$ | $< 1e^{-3}$ |
| RN44 | 0.18 | 0.138 | - | $< 1e^{-3}$ | 0.008 | 0.01 | 0.018 | $< 1e^{-3}$ |

Table 23: Statistical significance test of LayerAct functions on CIFAR100-C dataset.

**LA-SiLU**

|      | ReLU | LReLU | PReLU | SiLU | HardSiLU | Mish | GELU | ELU |
|------|------|-------|-------|------|----------|------|------|-----|
| RN20 | $< 1e^{-3}$ | $< 1e^{-3}$ | $< 1e^{-3}$ | $< 1e^{-3}$ | $< 1e^{-3}$ | $< 1e^{-3}$ | $< 1e^{-3}$ | $< 1e^{-3}$ |
| RN32 | $< 1e^{-3}$ | $< 1e^{-3}$ | $< 1e^{-3}$ | $< 1e^{-3}$ | $< 1e^{-3}$ | $< 1e^{-3}$ | $< 1e^{-3}$ | $< 1e^{-3}$ |
| RN44 | $< 1e^{-3}$ | $< 1e^{-3}$ | $< 1e^{-3}$ | $< 1e^{-3}$ | $< 1e^{-3}$ | $< 1e^{-3}$ | $< 1e^{-3}$ | $< 1e^{-3}$ |

**LA-HardSiLU**

|      | ReLU | LReLU | PReLU | SiLU | HardSiLU | Mish | GELU | ELU |
|------|------|-------|-------|------|----------|------|------|-----|
| RN20 | $< 1e^{-3}$ | $< 1e^{-3}$ | $< 1e^{-3}$ | $< 1e^{-3}$ | $< 1e^{-3}$ | $< 1e^{-3}$ | $< 1e^{-3}$ | $< 1e^{-3}$ |
| RN32 | $< 1e^{-3}$ | $< 1e^{-3}$ | $< 1e^{-3}$ | $< 1e^{-3}$ | $< 1e^{-3}$ | $< 1e^{-3}$ | $< 1e^{-3}$ | $< 1e^{-3}$ |
| RN44 | $< 1e^{-3}$ | $< 1e^{-3}$ | $< 1e^{-3}$ | $< 1e^{-3}$ | $< 1e^{-3}$ | $< 1e^{-3}$ | $< 1e^{-3}$ | $< 1e^{-3}$ |

