# OpenReview forum: "LayerAct: Advancing CNNs with BatchNorm through Layer-direction Normalization"
_ICLR.cc/2024/Conference — Submitted to ICLR 2024_

### Official Review · Reviewer_RDYZ · 2023-10-14

**Soundness:** 3 good
**Presentation:** 3 good
**Contribution:** 3 good
**Rating:** 8
**Confidence:** 2

**Summary:**

This paper introduces a novel activation mechanism for Convolutional Neural Networks (CNNs) with BatchNorm, addressing limitations of existing activation functions, specifically the trade-off problem and the large variance of noise-robustness across samples. The proposed LayerAct functions aim to provide layer-level activation, reducing noise fluctuations in activation outputs and achieving noise-robustness independently of the activation's saturation state. The authors present a comprehensive analysis and experimental results demonstrating the superiority of LayerAct functions over element-level activation functions in terms of noise-robustness. Additionally, they show that LayerAct functions perform exceptionally well in handling noisy datasets, outperforming element-level activation functions, while also achieving superior performance on clean datasets in most cases.

**Strengths:**

- The paper makes a significant contribution to the field of deep learning by introducing the concept of LayerAct functions, which address limitations in existing activation functions. This novel approach provides a valuable addition to the toolbox of techniques for improving the robustness and performance of CNNs.

- This paper is written in a clear and easily comprehensible manner, making it easy for readers to follow.

**Weaknesses:**

see Questions.

**Questions:**

- Some existing advanced batch normalization improvements like IEBN [1] and SwitchNorm [2] have shown enhanced performance. It would be interesting to investigate whether LayerAct can further improve the performance of these normalization methods.

- While I understand that this paper discusses CNNs, there is a growing need for advanced activation functions in various other network architectures, including transformers and UNets. I'd like to know if the proposed method has the potential to be adapted to these advanced network structures.

- I still don't quite understand the advantage of "layer-direction" activation over "element-wise" activation. Could the author please provide a concise explanation with simple examples or a summary?

- Additionally, I'd be interested in understanding in which applications LayerAct might excel or not excel. For example, we have instance norm for tasks like style transfer, batch norm for CNN-based classification tasks, and layer norm for transformer-related tasks. Can LayerAct be analyzed and discussed in a similar manner, suggesting suitable application areas?

- The author needs to clarify the above questions. If these issues are addressed, I will consider these clarifications along with feedback from other reviewers in deciding whether to raise my score.

[1] Instance Enhancement Batch Normalization: An Adaptive Regulator of Batch Noise, AAAI

[2] Differentiable Learning-to-Normalize via Switchable Normalization, ICLR

---

> ### Author Response · Authors · 2023-11-14
>
> ### Question 1. The potential of LayerAct to be adapted to advanced network structures.
> #### 1-1. The potential of LayerAct on transformers and networks with LayerNorm.
> - Discussing the interplay between LayerAct and LayerNorm is crucial for investigating LayerAct’s potential in transformer-based architectures, given LayerNorm's significant role in such networks. When LayerNorm is placed before the activation layer, it normalizes the mean and variance of the activation input. This pre-normalization results in the activation output of a LayerAct function being similar to that of the corresponding element-level activation function.
> - For example, consider a network using LayerNorm without an affine function (no gain and bias) and placing the LayerNorm layer right before the activation layer. Here, the normalization output of the LayerNorm layers serves as the activation input. Consequently, in this scenario, the normalization output of a LayerNorm layer is $n^{ \\hat{ LN } } = \\frac{ y -\\mu^2\_y }{\\sqrt{\\sigma^2\_y + \\alpha}}$. This leads to the output of SiLU $n^{ \\hat{ LN } } s\\left(n^{ \\hat{ LN } } \\right)$ and LA-SiLU $n^{ \\hat{ LN } } s \\left( n \\right)$ to be exactly same as $n = n^{\\hat{ LN }}\_i$.
> - As such, the closer the output of the LayerNorm layer approximates the normalized input of LayerAct’s activation scale function $s$ (i.e., the more gain and bias approximate to one and zero, respectively), the more the benefits of LayerAct are reduced.
> - Please note that LayerAct has benefits as an activation method, (LayerAct is not a normalization method): i) it addresses the trade-off problem between important properties of activation, and ii) has lower variance of noise-robustness across samples. These benefits can be maintained when used with LayerNorm that includes an affine function. In such scenarios, the layer-direction normalization in LayerNorm and LayerAct differs, represented as $n^{LN} = g \\cdot \\frac{ y -\\mu^2_y }{\\sqrt{\\sigma^2_y + \\alpha}} + b $ and $ n= \\frac{ y -\\mu^2_y }{\\sqrt{\\sigma^2_y + \\alpha}}$, respectively.
> - This implies that LayerAct can maintain its benefit when utilized with LayerNorm with an affine function, although it might be comparatively less than that on CNN-based networks with BatchNorm.
> - To validate this, we are conducting additional experiments with networks utilizing LayerNorm on CIFAR10 and CIFAR100. We will report the experimental results as soon as possible.
>
> #### 1-2. The potential of LayerAct on UNet.
> - LayerAct might have the potential to improve a UNet as it is a CNN-based architecture. However, we cannot be certain, as UNet is for the image segmentation task, while our experimental results were only on image classification tasks.
> - To address your concerns directly, we are conducting additional experiments with UNets on the brain image segmentation task. We will share the results of the experiment as soon as possible.
>
> ### Question 2. Investigation whether LayerAct can further improve the performance of other batch-direction normalization methods.
> - We agree that evaluating performance on networks with advanced normalization methods is important.
> - Considering our main contribution in this work is introducing a new activation method to enhance the performance of CNNs with BatchNorm, we plan to conduct additional experiments with ResNets with IEBN, decorrelated batch normalization (DBN; an advanced normalization method that utilizes batch-direction normalization introduced by Huang et al (2018)), and SwitchNorm. We will report the results of the remaining networks as soon as possible.
> - However, LayerAct’s effectiveness in networks using SwitchNorm is may be tempered, as SwitchNorm utilizes LayerNorm. This aligns with our earlier response, where we noted that LayerAct’s benefits might be reduced when LayerNorm precedes it.
>
> Huang, L., Yang, D., Lang, B., & Deng, J. “Decorrelated batch normalization.” CVPR. 2018.

---

> ### Author Response · Authors · 2023-11-14
>
> ### Question 3. The advantage of layer-level activation over element-level activation.
> - Thanks to the reviewers, we recognized the lack of a detailed discussion on one-side saturation of element-level activation functions, zero-like mean activation, and the trade-off problem between them. We will enhance our manuscript with a comprehensive explanation about them in the Appendix, based on this response.
> - In this response, we aim to clarify LayerAct’s contributions as an activation function. To this end, we will detail the limitations of existing element-level activation functions and how LayerAct can address those.
>
> #### 3-1. Summary of this response.
> - LayerAct functions are specifically designed to overcome the limitations of existing activation functions. These limitations include i) the trade-off problem between one-side saturation and zero-like mean activation, and ii) the large variance in noise-robustness across samples.
> - Notably, addressing the trade-off problem stands as a unique contribution of LayerAct. This benefit is unachievable with other activation functions, regardless of their integration with LayerNorm.
> - LayerAct has a lower upper bound of activation fluctuation compared to element-level activation functions with the same activation input vector. This is substantiated by our experimental analysis detailed in Subsection 4.1 of the manuscript and the experimental results on noisy image datasets.
>
> #### 3-2. Important properties of activation functions and the trade-off problem.
> - The saturation state of an activation function contributes to robustness against shifts in activation input, as the changes in saturated activation inputs minimally affect the output. However, early activation functions like Sigmoid and Tanh, which saturate both positive and negative sides, suffer from the vanishing gradient problem.
> - To overcome this while maintaining noise-robustness, one-sided saturation became an important property after the great success of ReLU. ReLU can ensure noise-robustness (as the outputs are not affected by the input shift in the saturation state) and avoids vanishing gradient problems by allowing large positive outputs.
> - However, functions like ReLU, which do not allow negative output, encounter another issue of bias shift, leading to ineffective and inefficient training. After Clevert et al. (2016) demonstrated that an activation mean closer to zero can solve such a problem, “zero-like mean activation” has become an important property of activation functions.
> - To incorporate both properties, one-sided saturation and zero-like mean activation, activation functions such as ELU, GELU, and SiLU were designed to allow small negative output to push the activation mean towards zero while saturating large negative output for noise-robustness.
> - Nevertheless, a problem still remains: there is a trade-off between one-sided saturation and zero-like mean activation. Saturating large negative output naturally restricts the zero-like mean activation. For example, allowing larger negative outputs pushes the activation mean closer to zero but diminishes the noise-robustness of the activation function as the range of the saturation state decreases.
> - Placing a LayerNorm layer parallel to an activation layer cannot address this trade-off problem. Large negative outputs will be restricted for saturation state. This is because the range of the saturation state entirely depends on the design of the activation function.
>
> #### 3-3. LayerAct can address the trade-off problem.
> - LayerAct functions address the trade-off between saturation and zero-like mean activation by integrating layer-direction normalization **into** the activation. The layer-direction normalization only determines the input of the activation scale function, which defines the saturation state (where $s\\left(n_{i}\\right)\\approx0)$.
> - This mechanism of LayerAct cuts off the link between saturation and negative activation output. Hence, LayerAct can produce larger negative output while maintaining saturation states. For example, in a scenario where $\\mu_{y}\\ll0$, the activation output $a_{i}=\\mu_{y}/2\\ll0$ where the activation input $y_{i}=\\mu_{y}$.
> - In conclusion, despite its structural simplicity, LayerAct functions offer a significant contribution by addressing the trade-off problem of activation.
>
> Clevert, Djork-Arné, Thomas Unterthiner, and Sepp Hochreiter. "Fast and Accurate Deep Network Learning by Exponential Linear Units (ELUS)." ICLR. 2016.

---

> ### Author Response · Authors · 2023-11-14
>
> #### 3-4. Element-level activation functions have large variance in noise-robustness across samples
> - The noise-robustness of element-level activation functions relies only on the saturation state. This implies that existing activation functions can ensure noise-robustness for samples only when a sufficiently large number of elements are in the saturation state, not when there are fewer elements in the saturation state.
> - For example, consider a ReLU layer. The samples with the lowest activation fluctuation are those where the activation input $y_{i}\\ll0$ for all $i$. Conversely, when the activation input $y_{i}\\gg0$ for all $i$, shifts in the activation input are directly transferred to the activation output.
> - We demonstrated this in Subsection 2.3 of our manuscript with activation fluctuation. Equation 4 shows that the upper bound of activation fluctuation is related to the activation scale, $\\| s\\left(\*\\right) \\|$ and $\\| s\\left(\\hat{\*} \\right) - s\\left(\*\\right) \\|$.
>
> #### 3-5. LayerAct has a lower upper bound of noise-robustness across samples compared to element-level activation.
> - As discussed in Subsection 3.2 with Equations 8 and 9 in our manuscript, LayerAct functions have a lower upper bound of activation fluctuation compared to element-level activation functions with the same activation input vector $y$. This results the LayerAct to have lower variance of noise-robustness compares to element-level activation functions across samples. Our experimental analysis on the distribution of activation fluctuation in Subsection 4.1 supports this.
> Clevert, Djork-Arné, Thomas Unterthiner, and Sepp Hochreiter. "Fast and Accurate Deep Network Learning by Exponential Linear Units (ELUS)." ICLR. 2016.
>
> ### Question 4. Application of LayerAct.
> - LayerAct has demonstrated its potential in image classification tasks using CNNs with BatchNorm, as evidenced by our experimental results on CIFAR10, CIFAR100, and ImageNet. Notably, the noise-robustness of LayerAct during inference is particularly advantageous for tasks where models have to be trained on clean datasets but need to classify or segment noisy images.
> - A relevant example is the classification of satellite images, which can be significantly affected by various external factors including weather conditions (Pritt, et al., 2017). Another example is task on motion blurred images in real-world due to the movement of the object or camera (Dai, 2008).
> - We would like to emphasize that the noisy datasets, which are utilized to evaluate the performance of the networks with LayerAct, include such real-world noises.
>
> Pritt, M., and Gary C. "Satellite image classification with deep learning.” IEEE applied imagery pattern recognition workshop (AIPR). 2017.
> Dai, S., & Wu, Y. “Motion from blur.” CVPR. 2008.
>
> #### Concluding remarks
> - We would like to reemphasize that LayerAct can enhance networks with BatchNorm through its ability to address the limitations of element-level activation functions. This unique contribution is unattainable through the parallel use of LayerNorm and element-level activation functions.
> - Once again, we sincerely appreciate your time and effort in reviewing our work. We have clarified the contribution, limitation and application of LayerAct, and have conducted additional experiments to enhance the precision and robustness of LayerAct. If there are any further concerns or questions, we are more than willing to address them.

---

> ### Author Response · Authors · 2023-11-22
>
> In this additional response, we present the results of additional experiments. We appreciate your comments that encouraged us to conduct valuable experiments.
>
> ### 1. LayerAct with UNet.
> - We agree that investigating the potential of LayerAct on different network architectures are important.
> - We conducted additional experiments on UNet and UNet++, which are CNN-based architectures for medical image segmentation tasks. For detailed analysis and results, we refer you to our Meta response 3.
> - The experimental results demonstrate that networks with LA-SiLU perform better than those with ReLU and SiLU in every instance. This highlights LayerAct’s practical potential in various CNN-based architectures and segmentation tasks.
>
> ### 2. ResNets with SwitchNorm, IEBN, and DBN.
> - To examine whether LayerAct can enhance the performance of networks with normalization methods other than BatchNorm, we trained ResNets with LayerNorm, SwitchNorm, IEBN, and DBN on the CIFAR10 and CIFAR100.
> - We used same experiment setting with those of our manuscript. We report the average accuracy over 10 runs for the experiments with SwitchNorm and IEBN, and average accuracy over 5 runs for those with DBN.
> - Analysis of the experimental results revealed that LayerAct functions are not effectively compatible with normalizations that can cause a large variance in the channel means, such as SwitchNorm and IEBN. This incompatibility arises because LayerAct is more sensitive to such large variance between channel means, which resulting channels with smaller means to be more likely to become inactivated compared to those with larger means.

---

> ### Author Response · Authors · 2023-11-22
>
> #### 2-1. ResNets with SwitchNorm.
> - SwitchNorm utilize the weighted average of three normalization method, BatchNorm, LayerNorm, and InstanceNorm.
>
> | Data | Model | Activation | Clean | Total | Noise | Blur | Digital | Weather | Extra |
> | - | - | - | - | - | - | - | - | - | - |
> | CIFAR10 | ResNet20 | ReLU |89.65 |72.4 |56.07 |71.35 |75.24 |80.43 |74.81 |
> | CIFAR10 | ResNet20 | SiLU |**90.6** |**74.17** |**57.7** |**72.94** |**77.57** |**82.21** |**76.33** |
> | CIFAR10 | ResNet20 | LA-SiLU |89.56 |72.43 |55.9 |71.05 |75.47 |80.87 |74.73 |
>
> - ResNet32 with SwitchNorm on CIFAR10.
>
> | Data | Model | Activation | Clean | Total | Noise | Blur | Digital | Weather | Extra |
> | - | - | - | - | - | - | - | - | - | - |
> | CIFAR10 | ResNet32 | ReLU |90.7 |73.9 |58.24 |72.84 |76.3 |81.79 |76.42 |
> | CIFAR10 | ResNet32 | SiLU |**90.79** |**74.65** |**58.78** |**73.39** |**77.64** |**82.57** |**76.91** |
> | CIFAR10 | ResNet32 | LA-SiLU |89.94 |72.72 |56.15 |71.07 |75.52 |81.48 |75.23 |
>
> - ResNet44 with SwitchNorm on CIFAR10.
>
> | Data | Model | Activation | Clean | Total | Noise | Blur | Digital | Weather | Extra |
> | - | - | - | - | - | - | - | - | - | - |
> | CIFAR10 | ResNet44 | ReLU |**91.4** |**74.65** |**58.1** |**74.01** |**77.18** |**82.81** |**76.99** |
> | CIFAR10 | ResNet44 | SiLU |74.48 |61.7 |49.5 |60.58 |64.11 |67.87 |63.41 |
> | CIFAR10 | ResNet44 | LA-SiLU |89.36 |72.02 |54.85 |70.59 |75.07 |81.05 |74.24 |
>
> - ResNet20 with SwitchNorm on CIFAR100.
>
> | Data | Model | Activation | Clean | Total | Noise | Blur | Digital | Weather | Extra |
> | - | - | - | - | - | - | - | - | - | - |
> | CIFAR100 | ResNet20 | ReLU |57.36 |37.01 |20.61 |38.43 |39.75 |43.56 |38.59 |
> | CIFAR100 | ResNet20 | SiLU |**64.55** |**42.96** |**25.13** |**43.91** |**46.35** |**50.42** |**44.55** |
> | CIFAR100 | ResNet20 | LA-SiLU |63.88 |41.76 |23.23 |42.83 |45.28 |49.43 |43.4 |
>
> - ResNet32 with SwitchNorm on CIFAR100.
>
> | Data | Model | Activation | Clean | Total | Noise | Blur | Digital | Weather | Extra |
> | - | - | - | - | - | - | - | - | - | - |
> | CIFAR100 | ResNet32 | ReLU |60.19 |38.81 |21.89 |40.02 |41.3 |46.0 |40.61 |
> | CIFAR100 | ResNet32 | SiLU |**64.35** |**42.45** |**25.00** |43.08 |**45.86** |49.92 |**44.00** |
> | CIFAR100 | ResNet32 | LA-SiLU |64.05 |42.27 |23.72 |**43.46** |45.5 |**50.16** |43.89 |
>
> - ResNet44 with SwitchNorm on CIFAR100.
>
> | Data | Model | Activation | Clean | Total | Noise | Blur | Digital | Weather | Extra |
> | - | - | - | - | - | - | - | - | - | - |
> | CIFAR100 | ResNet44 | ReLU |61.64 |39.93 |22.64 |41.03 |42.47 |47.54 |41.67 |
> | CIFAR100 | ResNet44 | SiLU |44.8 |29.57 |18.11 |29.96 |31.98 |34.21 |30.73 |
> | CIFAR100 | ResNet44 | LA-SiLU |**62.99** |**41.91** |**22.86** |**43.06** |**46.0**2 |**49.79** |**43.06** |
>
> - Our experiments exhibit that the combination of SwitchNorm and LayerAct is not effective.
> - Although further investigation is necessary to address why LayerAct is incompatible with SwitchNorm, our preliminary analysis is because LayerAct is more sensitive to the presence of similar channel characteristics across samples compared to element-level activation functions. If two samples display similar orders in the mean and variance of their channels, LayerAct, which employs layer-direction normalization for activation, tends to yield similar activation outputs.
> - BatchNorm distinctively ensure different channel characteristics between samples. While LayerNorm does not inherently differentiate channels across samples, it does not actively homogenize them either, thus preserving the natural order of mean and variance among channels. On the other hand, InstanceNorm actively normalizes the mean and variance of channels to be more uniform.
> - Consequently, due to the integrated normalization approach of SwitchNorm, which combines three methods, we expect that the effect of InstanceNorm tends to homogenize channel characteristics, rendering LayerAct functions susceptible to inefficiency.

---

> > ### Author Response · Authors · 2023-11-22
> >
> > #### 2-2. LayerAct with IEBN
> > - IEBN utilize instance-direction average $m\_{b c}$ during calculating $\\delta\_{bc} = sig \\left( \\hat{ \\gamma }\_{c} \\cdot m\_{b c} + \\hat{ \\beta }\_c  \\right)$. Here, $\\delta\_{bc}$ operates as a weight coefficient for each channel in data points.
> >
> > - ResNet20 with IEBN on CIFAR10.
> >
> > | Data | Model | Activation | Clean | Total | Noise | Blur | Digital | Weather | Extra |
> > | - | - | - | - | - | - | - | - | - | - |
> > | CIFAR10 | ResNet20 | ReLU |91.5 |70.12 |52.49 |66.73 |73.44 |79.64 |73.89 |
> > | CIFAR10 | ResNet20 | SiLU |91.44 |68.98 |49.39 |66.26 |72.29 |79.0 |73.06 |
> > | CIFAR10 | ResNet20 | LA-SiLU |**91.63** |**70.64** |**52.7** |**67.3** |**74.2** |**79.87** |**74.65** |
> >
> > - ResNet32 with IEBN on CIFAR10.
> >
> > | Data | Model | Activation | Clean | Total | Noise | Blur | Digital | Weather | Extra |
> > | - | - | - | - | - | - | - | - | - | - |
> > | CIFAR10 | ResNet32 | ReLU |**92.54** |71.67 |54.06 |**68.82** |74.82 |81.12 |75.11 |
> > | CIFAR10 | ResNet32 | SiLU |91.85 |71.07 |53.39 |68.26 |73.9 |80.5 |74.9 |
> > | CIFAR10 | ResNet32 | LA-SiLU |92.36 |**71.77** |**54.19** |68.19 |**75.32** |**81.15** |**75.6** |
> >
> > - ResNet20 with IEBN on CIFAR10.
> >
> > | Data | Model | Activation | Clean | Total | Noise | Blur | Digital | Weather | Extra |
> > | - | - | - | - | - | - | - | - | - | - |
> > | CIFAR10 | ResNet44 | ReLU |**92.78** |72.18 |55.2 |68.83 |75.61 |81.59 |75.43 |
> > | CIFAR10 | ResNet44 | SiLU |92.08 |71.23 |52.79 |68.54 |74.27 |80.95 |74.99 |
> > | CIFAR10 | ResNet44 | LA-SiLU |92.5 |**73.01** |**56.97** |**69.18** |**76.03** |**82.34** |**76.5** |
> >
> > - ResNet20 with IEBN on CIFAR100
> >
> > | Data | Model | Activation | Clean | Total | Noise | Blur | Digital | Weather | Extra |
> > | - | - | - | - | - | - | - | - | - | - |
> > | CIFAR100 | ResNet20 | ReLU |66.62 |**41.97** |**22.81** |**42.51** |45.07 |50.3 |44.37 |
> > | CIFAR100 | ResNet20 | SiLU |66.19 |40.88 |21.14 |41.08 |44.29 |49.68 |43.28 |
> > | CIFAR100 | ResNet20 | LA-SiLU |**66.73** |41.59 |21.31 |41.98 |**45.26** |**50.53** |**43.82** |
> >
> > - ResNet32 with IEBN on CIFAR100
> >
> > | Data | Model | Activation | Clean | Total | Noise | Blur | Digital | Weather | Extra |
> > | - | - | - | - | - | - | - | - | - | - |
> > | CIFAR100 | ResNet32 | ReLU |**68.17** |43.49 |23.81 |**43.89** |46.76 |52.3 |**45.78** |
> > | CIFAR100 | ResNet32 | SiLU |67.43 |42.3 |23.15 |42.38 |45.33 |51.14 |44.68 |
> > | CIFAR100 | ResNet32 | LA-SiLU |67.97 |**43.56** |**24.49** |43.39 |**46.94** |**52.47** |45.76 |
> >
> > - ResNet44 with IEBN on CIFAR100
> >
> > | Data | Model | Activation | Clean | Total | Noise | Blur | Digital | Weather | Extra |
> > | - | - | - | - | - | - | - | - | - | - |
> > | CIFAR100 | ResNet44 | ReLU |**69.47** |44.73 |25.74 |44.77 |47.77 |53.51 |47.12 |
> > | CIFAR100 | ResNet44 | SiLU |68.18 |43.47 |24.76 |43.37 |46.46 |52.22 |45.89 |
> > | CIFAR100 | ResNet44 | LA-SiLU |68.41 |**45.29** |**26.89** |**45.15** |**48.67** |**54.03** |**47.13** |
> >
> > - With the exception of ResNet20 on CIFAR100, networks with LA-SiLU demonstrated enhanced performance on noisy datasets when compared to their counterparts utilizing ReLU and SiLU.
> > - Conversely, ReLU outperformed LA-SiLU in ResNet32 and ResNet44 models. Nonetheless, it is important to highlight that LA-SiLU consistently surpassed SiLU, which utilizes the same activation scale function, across all tested scenarios.
> > - These results imply that the mechanism of LayerAct holds promise for enhancing efficiency. It is also important to consider careful consideration and integration of network complexity, particularly due to the interplay between normalization and activation functions, are essential, given that the scale function of ReLU is considerably simpler compared to sigmoid, the scale function of LA-SiLU and SiLU.

---

> > > ### Author Response · Authors · 2023-11-22
> > >
> > > #### 2-3. LayerAct with DBN
> > > - DBN employs ZCA whitening as its normalization mechanism. This approach achieves the benefit of decorrelation while avoiding the “stochastic axis swapping issue” often encountered in similar processes, PCA-whitening.
> > >
> > > - ResNet20 with DBN on CIFAR10.
> > >
> > > | Data | Model | Activation | Clean | Total | Noise | Blur | Digital | Weather | Extra |
> > > | - | - | - | - | - | - | - | - | - | - |
> > > | CIFAR10 | ResNet20 | ReLU |89.96 |64.9 |35.84 |64.19 |71.27 |77.93 |67.99 |
> > > | CIFAR10 | ResNet20 | SiLU |89.69 |65.02 |39.63 |63.08 |70.24 |76.98 |68.83 |
> > > | CIFAR10 | ResNet20 | LA-SiLU |**91.33** |**68.49** |**43.02** |**67.59** |**73.41** |**79.73** |**72.33** |
> > >
> > > - ResNet32 with DBN on CIFAR10.
> > >
> > > | Data | Model | Activation | Clean | Total | Noise | Blur | Digital | Weather | Extra |
> > > | - | - | - | - | - | - | - | - | - | - |
> > > | CIFAR10 | ResNet32 | ReLU |91.99 |68.18 |37.10 |68.58 |**75.0** |81.30 |71.15 |
> > > | CIFAR10 | ResNet32 | SiLU |**92.27** |68.02 |40.16 |66.64 |73.86 |80.91 |71.57 |
> > > | CIFAR10 | ResNet32 | LA-SiLU |92.26 |**70.33** |**44.43** |**69.65** |**75.0** |**82.12** |**73.99** |
> > >
> > > - ResNet44 with DBN on CIFAR10.
> > >
> > > | Data | Model | Activation | Clean | Total | Noise | Blur | Digital | Weather | Extra |
> > > | - | - | - | - | - | - | - | - | - | - |
> > > | CIFAR10 | ResNet44 | ReLU |92.30 |69.07 |38.99 |69.20 |**75.73** |81.82 |72.07 |
> > > | CIFAR10 | ResNet44 | SiLU |92.13 |68.89 |41.72 |68.09 |74.14 |81.17 |72.52 |
> > > | CIFAR10 | ResNet44 | LA-SiLU |**92.42** |**70.94** |**46.53** |**69.58** |75.17 |**82.61** |**74.69** |
> > >
> > > - ResNet20 with DBN on CIFAR100
> > >
> > > | Data | Model | Activation | Clean | Total | Noise | Blur | Digital | Weather | Extra |
> > > | - | - | - | - | - | - | - | - | - | - |
> > > | CIFAR100 | ResNet20 | ReLU |**59.71** |**34.19** |13.48 |**34.26** |**38.28** |**43.67** |**36.07** |
> > > | CIFAR100 | ResNet20 | SiLU |57.82 |32.13 |13.41 |31.57 |35.78 |40.99 |34.22 |
> > > | CIFAR100 | ResNet20 | LA-SiLU |58.19 |32.24 |**14.32** |30.95 |35.11 |41.4 |34.96 |
> > >
> > > - ResNet32 with DBN on CIFAR100
> > >
> > > | Data | Model | Activation | Clean | Total | Noise | Blur | Digital | Weather | Extra |
> > > | - | - | - | - | - | - | - | - | - | - |
> > > | CIFAR100 | ResNet32 | ReLU |54.06 |30.29 |14.16 |28.67 |34.09 |38.34 |32.14 |
> > > | CIFAR100 | ResNet32 | SiLU |62.0 |35.26 |14.99 |34.41 |39.35 |45.01 |37.46 |
> > > | CIFAR100 | ResNet32 | LA-SiLU |**65.02** |**38.75** |**19.18** |**37.84** |**41.81** |**48.66** |**41.36** |
> > >
> > > - ResNet44 with DBN on CIFAR100
> > >
> > > | Data | Model | Activation | Clean | Total | Noise | Blur | Digital | Weather | Extra |
> > > | - | - | - | - | - | - | - | - | - | - |
> > > | CIFAR100 | ResNet44 | ReLU |60.32 |34.64 |14.96 |33.68 |39.03 |44.1 |36.51 |
> > > | CIFAR100 | ResNet44 | SiLU |64.67 |37.58 |15.81 |37.63 |42.01 |47.2 |39.83 |
> > > | CIFAR100 | ResNet44 | LA-SiLU |**67.59** |**42.44** |**21.03** |**42.54** |**46.52** |**52.15** |**44.6** |
> > >
> > > - Except for ResNet20 on CIFAR100, networks with LA-SiLU showed similar or improved performance on both clean and noisy datasets compared to those employing ReLU and LA-SiLU.
> > > - The results of these experiments highlight the potential applicability of LayerAct functions in conjunction with advanced batch-direction normalization methods.
> > >
> > > ### Concluding remark.
> > > - Additional experiments on networks with and without various normalization methods have revealed that the efficacy of LayerAct functions is significantly influenced by the choice of normalization method.
> > > - While LayerNorm can reduce the benefit of LayerAct, there is an empirical finding suggest that InstanceNorm could negatively impact LayerAct's performance. These results underscore the importance of batch-direction normalization for the optimal implementation of LayerAct.
> > > - The other additional experiments on UNet and UNet++ demonstrate that LayerAct functions have potential in enhancing performance across different architectural frameworks and in various applications, such as image segmentation.
> > > - Once again, we appreciate your insightful comments that inspired these additional experiments. Your observation not only reveals the potential of LayerAct functions across different architectures and tasks but also highlight the importance of exploring the relationship between normalization methods and LayerAct. We will add a discussion section in our manuscript based on our responses of your comments. If there are any further concerns or questions, we are more than willing to address them.

---

> ### Comment · Reviewer_RDYZ · 2023-11-23
> **Thank you for you response.**
>
> Thank you for your detailed response; my concerns have been addressed. I lean towards improving my scores. Additionally, I suggest that you can now include the relevant citations and additional experiments in the revised pdf (highlighted, for example, in blue font).. Thank you for your efforts.

---

> > ### Author Response · Authors · 2023-11-23
> >
> > We are pleased to know that our responses have adequately addressed your concerns. We deeply appreciate your insightful comments. Your suggestions guided us to conduct valuable experiments. The findings from these experiments have significantly enhanced the quality of our work.
> >
> > We are currently in the process of revising the manuscript and will ensure its timely submission. We extend our heartful thanks for the time and effort you have dedicated to reviewing our paper.

---

### Official Review · Reviewer_1Eep · 2023-10-31

**Soundness:** 2 fair
**Presentation:** 3 good
**Contribution:** 2 fair
**Rating:** 5
**Confidence:** 4

**Summary:**

This paper proposes an activation mechanism called LayerAct that combines layer normalization in the (general) sigmoid-linear-units to improve the noise-robust of the network. It empirically shows that the proposed LayerAct functions have a zero-like mean activation and are more noise-robustness. Experimental results with three clean and three out-of-distribution benchmark datasets for image classification tasks show the proposed LayerAct functions output perform the baselines on noisy datasets, and also is also superior on clean datasets in most cases.

**Strengths:**

1. The topic is interesting and important for the community.
2. The proposed LayerAct can improve the performance marginally over the baselines, showing the potential in practice.
3. It is glad to see the proposed LayerAct exhibit superior noise-robustness compared to element-level activation functions.

**Weaknesses:**

1. I think the technical contribution is overall low. The proposed LayerAct can be viewed as Layer Normalization combining the (general) sigmoid-linear-units. Both of them are good methods in improving the performance of neural network. It is not surprise the proposed method can improve the performance over the original activation functions. It is good that this paper addresses the robust of the proposed method, however I have concerns in the second point.

2. I have concerns on the clarity of why LayerAct is noise-robust. The main claim is that the scale activation function is bounded (Eqn.8 and Eqn.9), thus the model is robust. But, bounding the activation function cannot bounding the $y_i$, which can still be not robust. This paper should well address this point. It is true the proposed LayerAct can obtain good empirical results on corruption datasets. However, this is not surprising, because the previous methods[1] have show that the combination of Batch-Free normalization (e.g., LayerNorm) and BatchNorm can be more robust for distribution shift (e.g., corruption). The main insights is that LayerNorm can alleviate the train-inference inconsistency problem of BatchNorm (I noted all experiments on noise-robust uses the networks with BatchNorm). I want to ask whether the proposed LayerAct can be noise-robust on the network without BatchNorm?

3. I have concern on the title of this paper. The title addresses “ADVANCING CNNS WITH BATCHNORM THROUGH LAYER-DIRECTION NORMALIZATION”. However, I find the description of the proposed LayerAct is independent to the BatchNorm (e.g, this paper doesnot say how LayerAct alleviates the problem of CNN with BatchNorm). I think this paper should well clarify it

4. This paper has some imprecise descriptions:
(1)I have concerns on this “Specifically, we propose a novel layer-level activation (LayerAct) mechanism, along with two as sociated functions. This advancement combines batch-direction normalization with the effects of layer-direction normalization”. How does LayerAct combines both? If yes, why the experiments run on the network with BatchNorm? How does the LayerAct works on the network without BatchNorm.
(2) This paper claims “One-sided saturation avoids the vanishing gradient problem while maintaining noise-robustness”. Why the One-sided saturation avoids the vanishing gradient problem? Based on my understanding, the saturation state will cause no gradient.
(3)why ” the sum of activation scale $\|s(n^{LN})\|$ will be similar across all samples”?  please clarify it in detail.


Other minors:
“pay cloase attention” in page 1.


**Ref:**
[1] Delving into the Estimation Shift of Batch Normalization in a Network. CVPR 2022

**Questions:**

see weaknesses

---

> ### Author Response · Authors · 2023-11-14
>
> Thank you very much for your comments. We are glad to see that our topic and experiments meet your preference. In response to your concerns and questions, we describe our responses as follows.
>
> ### Weakness 1. I think the technical contribution is overall low. & Weakness 4-2. About the statement on one-sided saturation.
> - Thanks to the reviewers, we recognized the lack of detailed discussion on one-sided saturation, zero-like mean activation, and the trade-off problem between them. We will enhance our manuscript with a comprehensive explanation about them in the Appendix, based on the following response.
> - In this response, we aim to clarify LayerAct's contributions as a new activation mechanism. To this end, we will detail the limitations of existing element-level activation functions that cannot be addressed by the mere combination of LayerNorm and activation functions. This response also resolves the concerns raised about one-sided saturation in Weakness 4 (See 1-2 of this response).
>
> #### 1-1. Summary of this response.
> - We acknowledge that the structure of LayerAct, a combination of LayerNorm and activation functions, is simple. Behind this simple mechanism, it is important to emphasize that LayerAct offers unique benefits that cannot be achieved through utilizing LayerNorm and element-level activation layers together.
> - LayerAct functions specifically address the limitations of existing activation functions: i) the trade-off problem between one-sided saturation and zero-like mean activation, and ii) the large variance of noise-robustness across samples. In this response, we focus on the first benefit, as the significance of noise-robustness is already well recognized (For the response to Weakness 2 about the noise-robustness of LayerAct, please see the next response).
> - Notably, addressing the trade-off problem stands as a unique contribution of LayerAct. This benefit is unachievable with other activation functions, regardless of their integration with LayerNorm.
>
> #### 1-2. Important properties of activation functions and the trade-off problem.
> - The saturation state of an activation function contributes to robustness against shifts in activation input, as the changes in saturated activation inputs minimally affect the output. However, early activation functions like Sigmoid and Tanh, which saturate both positive and negative sides, suffer from the vanishing gradient problem.
> - To overcome this while maintaining noise-robustness, one-sided saturation became an important property after the great success of ReLU. ReLU can ensure noise-robustness (as the outputs are not affected by the input shift in the saturation state) and avoids vanishing gradient problems by allowing large positive outputs.
> - However, functions like ReLU, which do not allow negative output, encounter another issue of bias shift, leading to ineffective and inefficient training. After Clevert et al. (2016) demonstrated that an activation mean closer to zero can solve such a problem, “zero-like mean activation” has become an important property of activation functions.
> - To incorporate both properties, one-sided saturation and zero-like mean activation, activation functions such as ELU, GELU, and SiLU were designed to allow small negative output to push the activation mean towards zero while saturating large negative output for noise-robustness.
> - Nevertheless, a problem still remains: there is a trade-off between one-sided saturation and zero-like mean activation. Saturating large negative output naturally restricts the zero-like mean activation. For example, allowing larger negative outputs pushes the activation mean closer to zero but diminishes the noise-robustness of the activation function as the range of the saturation state decreases.
> - Placing a LayerNorm layer parallel to an activation layer cannot address this trade-off problem. Large negative outputs will be restricted for saturation state. This is because the range of the saturation state entirely depends on the design of the activation function.
>
> Clevert, Djork-Arné, Thomas Unterthiner, and Sepp Hochreiter. "Fast and Accurate Deep Network Learning by Exponential Linear Units (ELUS)." ICLR. 2016.

---

> ### Author Response · Authors · 2023-11-14
>
> #### 1-3. LayerAct can address the trade-off problem.
> - LayerAct functions address the trade-off between saturation and zero-like mean activation by integrating layer-direction normalization **into** the activation. The layer-direction normalization only determines the input of the activation scale function, which defines the saturation state (where $s\\left(n_{i}\\right)\\approx0)$.
> - This mechanism of LayerAct cuts off the link between saturation and negative activation output. Hence, LayerAct can produce larger negative output while maintaining saturation states. For example, in a scenario where $\\mu_{y}\\ll0$, the activation output $a_{i}=\\mu_{y}/2\\ll0$ where the activation input $y_{i}=\\mu_{y}$.
> - In conclusion, despite its structural simplicity, LayerAct functions offer a significant contribution by addressing the trade-off problem of activation.
>
>
> ### Weakness 2. Concerns on the noise-robustness of LayerAct.
> #### 2-1. The activation input $y_{i}$ is not bounded, which impact noise-robustness significantly.
> - We agree that the activation input $y_{i}$ significantly influences activation fluctuation. However, it is important to note that this challenge applies to all activation functions, not only to LayerAct functions.
> - Our objective was to demonstrate that LayerAct functions have smaller activation fluctuations **compared to** other activation functions under identical conditions, in our case, the same activation input.
> - We recognize that our explanation in Subsection 3.2 of the submitted manuscript has not been sufficiently clear. Our objective was to demonstrate that LayerAct functions have smaller activation fluctuations **compared to** other activation functions under identical conditions, in our case, the same activation input.
> - Comparing methods under identical conditions has been used to analyze the effects of a method. Santurkar et al. (2018) compared networks with and without BatchNorm under the same gradient conditions to assess BatchNorm’s impact on optimization. Similarly, Xu et al. (2019) conducted comparisons of different variants of LayerNorm under the same input conditions.
> - In this way, we showed that LayerAct functions can ensure more robust processing during the forward pass compared to other activation functions, given the same level of activation input. This benefit of LayerAct is supported by both mathematical analysis (in Subsection 3.2) and experimental results (in Subsection 4.1).
> - Thus, unless the training process does not lead to the generation of larger activation inputs in networks utilizing LayerAct, LayerAct is noise-robust compared to element-level activation functions. Please refer to Figure 7 in Appendix E, which demonstrates the distribution of activation input across activation functions. The figure shows that LayerAct functions typically maintain a similar or lower mean of activation inputs.
> - In summary, LayerAct functions are designed to facilitate more robust processing during the forward pass, as was the primary aim of Equations 8 and 9. We acknowledge the need for clarity in our manuscript on this matter and will make revisions to eliminate any confusion.
> Santurkar, S., Tsipras, D., Ilyas, A., and Madry, A. “How does batch normalization help optimization?”. Neurips. 2018
> Xu, J., Sun, X., Zhang, Z., Zhao, G., and Lin, J. “Understanding and improving layer normalization.” Neurips. 2019
>
> #### 2-2. Does the noise-robustness of LayerAct maintain without BatchNorm?
> - It is important to highlight that the activation input $y_{i}$ in Equations 8 and 9 is not exclusively derived from the output of a normalization layer. Additionally, in the experimental analysis detailed in Subsection 4.1, we did not use any normalization methods. The results from these experiments affirm that the noise-robustness of LayerAct is not contingent on the use of normalization layers.
> - To directly address your concern, we are conducting additional experiments using ResNets without any normalization layers. We have demonstrated the experimental results in our global response. In cases of networks without any normalization method, certain trials of ResNet20 and 44 with SiLU exploded during training. Consequently, we are presenting the result from networks with ReLU and LA-SiLU.
> - On clean datasets, the performance varied with the complexity of the networks: ResNet 32 and 44 with ReLU demonstrated better performance compared to those with LA-SiLU on both CIFAR10 and CIFAR100, whereas the opposite was true for ResNet.
> - Here, it is important to observe that networks with LA-SiLU outperformed those with ReLU on noisy datasets in all cases, which reconfirm the intended utility of LayerAct mechanism and contribution of our work. This experimental result shows that the noise-robustness of LayerAct is independent of the usage of normalization.

---

> ### Author Response · Authors · 2023-11-14
>
> ### Weakness 3. Concerns on the title of the paper.
> - Firstly, we wish to clarify that our intention was to present our contributions in a conservative manner. Considering that CNNs primarily utilize BatchNorm in their normalization layers, and our experimental results are based on ResNets with BatchNorm, the purpose of our title was to underscore the unique benefits of LayerAct as an activation mechanism in enhancing the performance of CNNs that utilize BatchNorm.
> - However, we acknowledge that the current title may inadvertently imply a direct association between LayerAct and BatchNorm. To eliminate any potential confusion, we have decided to revise the title to:
> >- “LayerAct: Advanced activation mechanism utilizing layer-direction normalization for CNNs with BatchNorm”
> - If there are still concerns that the revised title might lead to the same confusion, another option could be to remove the specific reference to "BatchNorm.":
> >- “LayerAct: Advanced activation mechanism utilizing layer-direction normalization for CNNs”
>
> ### Weakness 4. Imprecise descriptions.
> #### 4-1. About the statement “combines batch-direction normalization with the effects of layer-direction normalization”
> - In line with our previous response, we acknowledge that the statement may cause confusion. To provide greater clarity, we have revised the statement from “This advancement combines batch-direction normalization with the effects of layer-direction normalization, yielding two benefits: i) addressing the trade-off issue between two significant properties of activation, and ii) improving the noise-robustness of networks by reducing the variance of noise-robustness among samples.“ to the following:
> >- “While maintaining the batch-direction normalization methods, which are effective and prevalent in CNN-based networks, this advancement can provide the benefits of layer-direction normalization. This integration offers two main advantages: i) addressing the trade-off issue between two significant properties of activation, and ii) improving the noise-robustness of activations by reducing the variance of noise-robustness across samples.”
>
> #### 4-2. About the statement on one-sided saturation.
> - Please refer to the “1-2. Important Properties of Activation Functions and the Trade-off Problem.” in our response to Weakness 1.
>
> #### 4-3. About the statement on the sum of activation scale.
> - We recognize that the statement is only true when all of the activation inputs across samples are well distributed. Therefore, we will revise the sentence “With LayerNorm, the sum of activation scale $||s\left(n^{LN}\right)||$ will be similar across all samples, which helps to reduce the variance of noise-robustness across all samples” to:
> >- “With LayerNorm, the sum of activation scale $||s\left(n^{LN}\right)||$ will have a lower upper bound across all samples, which helps to reduce the variance of noise-robustness across all samples.”
>
> #### Concluding remarks.
> - We would like to reemphasize that the main contribution of LayerAct is its ability to address the limitations of element-level activation functions. This unique contribution is unattainable through the parallel use of LayerNorm and element-level activation functions and remains effective independent of BatchNorm.
> - Once again, we sincerely appreciate your time and effort in reviewing our work. We have been able to significantly enhance the presentation of our work, including revising the title and clarifying the contribution of LayerAct. If there are any further concerns or questions, we are more than willing to address them.

---

> > ### Comment · Reviewer_1Eep · 2023-11-19
> > **Reply to the authors responses.**
> >
> > I thank for the authors' responses in detail. I have read the responses.
> >
> > 1.	It is appreciated that the authors provide additional experiments on CNN (ResNet) without BatchNorm. It seems true that LayerAct can improve noise-robustness empirically. However, my concerns on the clarity of why LayerAct is noise-robust still hold, ie, I still think the analyses in Subsection 3.2 is not well supported by evidences. As I stated before, “bounding the activation function cannot bounding the $y_i$, which can still be not robust.”. If the analyses in bounding is correct, does this mean we can use ReLU-6 (the maximum value is 6, or other bounded activations) to obtain good noise-robustness? Besides, I want to know how the authors calculate the mean/variance for the LayerNorm-style operation in LayerAct for the convolutional input? E.g., for an convolutional input $X \in \mathbb{R}^{d \times h \times w}$, which dimensions the mean/variance is calculated over?
> > I also noted that LayerAct works worse than ReLU on ResNet20/44/56 without BatchNorm in terms of the clean results from the additional experiments, why the results are not consistent with the results on ResNet20/44/56 with BatchNorm? From these results, it seems that the performances of LayerAct are related to the network with/without BatchNorm. This further amplified my concerns 3 in the initial review, that is, this paper seems to be related to BatchNorm (as stated in the Title), but why the description of the proposed LayerAct is independent to the BatchNorm. In another view, this paper shows many words on BatchNorm in the first paragraph of the introduction, why this paper says the main aim is to design an activation function, but not related to BatchNorm in the response? I think this paper should provide a major revision in the presentation.
> >
> > 2.	I still have the concerns on “ the One-sided saturation avoids the vanishing gradient problem.” in the description. Let me make it indetail:(1) if one use ReLU (One-sided saturation activation), and many neurons are not activated (in the saturated state). In this case, I think the model will suffer vanishing gradients; (2) If one use ReLU and the weight is initialized with very small variance. In this case,  I think the model will suffer vanishing gradients. Therefore, I am not convinced to this sentence “One-sided saturation avoids the vanishing gradient problem"

---

> ### Author Response · Authors · 2023-11-21
>
> We appreciate your second comment. To address the remain concerns, our responses are as follows.
>
> ----
>
> ### Noise-robustness of LayerAct
> - We are pleased that the contribution of LayerAct's noise-robustness has been empirically recognized through our additional experiment.
> - We acknowledge that LayerAct may not maintain robustness when $y_{i}$ is large, as it does not bound it. This realization highlights the importance of utilizing a normalization method in networks with LayerAct functions. We plan to explicitly emphasize the necessity of using normalization methods to maintain the robustness of LayerAct in our manuscript.
> - The insights provided have led us to reevaluate our usage of the term “ensure” in certain contexts in our manuscript, recognizing that it may not fully capture all scenarios. To address this, we will revise the statement, “This suggests that LayerAct can ensure more robust processing during the forward pass of a network.” in our manuscript.
> >- This implies that networks with LayerAct are likely to achieve more robust processing during the forward pass, especially when the input $y$ is not excessively large, reinforcing the importance of applying normalization methods, such as BatchNorm, in networks with LayerAct functions.
>
> ----
>
> ### Noise-robustness of activation functions with saturation state.
> - The other point of your concern seems to be about all activation functions with a saturation state. We understand that your comment means, "Activation functions with a saturation state cannot ensure noise-robustness."
> - As your observations, one-sided saturation does not universally guarantee noise-robustness across all samples. Activation functions with a saturation state only provide noise-robustness for those samples that have a sufficient number of elements in the saturation state. This does not apply to samples with large $y_i$ for all $i$. However, please note that this observation highlights the limitation of activation functions that we criticized, namely the large variance in noise-robustness across samples.
> - Again, your insights have led us to recognize that our use of the term “ensure” in certain statements may not have adequately accounted for such instances.
> - Our intention was to claim that activation functions with a saturation state **potentially offer greater noise-robustness in more samples** compared to those without any saturation state. This is under the condition that a sufficiently large number of elements are in the saturation state.
> - For example, ReLU can provide noise-robustness for the samples when a sufficient number of elements are in the saturation state, indicating enough elements with small $y_i$. Conversely, activation functions without any saturation state, such as PReLU, do not provide noise-robustness for samples with small $y_i$. In this context, Clevert et al. (2016) stated, “In contrast to ReLUs, activation functions like LReLUs, PReLUs, and RReLUs do not ensure a noise-robust deactivation state.”
> - Our experimental results also corroborate the relationship between noise-robustness and saturation state. In Tables 2 and 3, the performance of networks with ReLU is better than those with PReLU on noisy datasets.
> - To clarify our statements, we will revise the statement in our manuscript, “This implies that existing activation functions can ensure noise-robustness for samples only when a sufficiently large number of elements are in the saturation state, not when there are fewer elements in the saturation state.”
> >- This implies that existing activation functions with a saturation state are **expected to be robust** for samples only when a sufficiently large number of elements are in the saturation state, not when there are fewer elements in the saturation state.
> - In summary, while no activation function can guarantee noise-robustness for samples with large $y_i$ with their saturation state, those with a saturation state are expected to be robust for a greater number of samples compared to those without a saturation state.
>
> ----
>
> ### The dimensions of mean/variance calculation.
> - In case of image data, the term $d$ of the equations in Subsection 2.3 and 3.2 denotes the representation of these dimensions as a single flattened vector. Thus, considering $X$ with dimension $c \\times w \\times h$, where $c$ is the channel dimension, $d=c \\times w \\times h$. The equation of mean $\\mu_y$ and variance $\\sigma_y^2$ are as follows:
> $$
> \\mu_y=\frac{1}{c \\times h \\times w}\\sum^c_{i=1} \\sum^h_{j=1} \\sum^w_{k=1} x_{i, j, k} = \\frac{1}{d}\\sum^d_{i=1} x_i
> $$
> $$
> \\sigma_y^2=\frac{1}{c \\times h \\times w}\\sum^c_{i=1} \\sum^h_{j=1} \\sum^w_{k=1} \\left(x_{i, j, k} - \\mu_y \\right)^2 = \\frac{1}{d}\\sum^d_{i=1} \\left(x_i-\\mu_y\\right)^2
> $$
> where $x_{c, h, w}$ and $x_{d}$ is a pixel of the image $X$.

---

> ### Author Response · Authors · 2023-11-21
>
> ### Relationship between LayerAct and BatchNorm.
> - There was a misinterpretation of your questions in our initial response. Our intention was to assert that LayerAct does not **help** BatchNorm.
> - We would like to reemphasize that we tried to present our contribution in a conservative manner. The title and introduction of our manuscript focused on the CNNs that utilize BatchNorm, since all the networks of experiments utilize BatchNorm.
> - However, we acknowledge that there are no detailed discussion of the relationship between LayerAct and BatchNorm. In this response, we will explore the relationship between LayerAct and BatchNorm.
> - LayerAct **needs** BatchNorm. Aforementioned, networks with LayerAct functions need to utilize normalization methods to maintain the benefit of noise-robustness. Another necessity of BatchNorm arises because LayerAct functions are complex activation functions that tend to cause networks to overfit the training datasets.
> - We will revise our manuscript to underscore the necessity of integrating BatchNorm with LayerAct. Additionally, a new section will be added to the appendix, providing detailed discussion into this requirement.
>
> | data | model | ReLU | LA-SiLU |
> | - | - | - | - |
> | CIFAR100 | ResNet32 | 92.23 | 94.52 |
> | CIFAR100 | ResNet44 | 95.76 | 96.47 |
>
> - To empirically investigate why ResNet32 and ResNet44 with LA-SiLU underperform compared to those with ReLU, especially when no normalization method is employed, we conducted inference tests on the trained networks without a normalization method using **the training dataset**. The results of table above show that networks with LA-SiLU exhibit higher accuracy than those with ReLU, suggesting a tendency toward overfitting.
> - Considering that LA-SiLU demonstrates better performance on the shallower network, ResNet20, compared to ReLU, these experimental results suggest that LA-SiLU requires modules that can prevent overfitting, such as normalization methods, for optimal performance in deep networks.
> - In summary, networks with LayerAct functions need BatchNorm to maintain the benefit of LayerAct’s noise-robustness and prevent the overfitting.
>
> ----
>
> ### About one-side saturation and vanishing gradient problem
> - Similar to our response on the relationship between noise-robustness and one-sided saturation, we acknowledge that the term “avoid” in the statement “avoid the vanishing gradient problem” is overly absolute, given the existence of counterexamples.
> - Our intention was to convey that activation functions with one-sided saturation have the potential to facilitate more effective propagation during the backward pass, which can alleviate the vanishing gradient problem. This potential arises from their partially larger derivative compared to functions with both-sided saturation, like sigmoid or tanh. For example, the derivative of ReLU is one on the positive side, while the maximum derivative of sigmoid is 0.25.
> - Previous studies have highlighted the attribute of ReLU, which has a derivative of one on its positive side, leading to more informative processing during the backward pass (Glorot et al., 2011; Clevert et al., 2016; Ramachandran et al., 2018).
> - Thus, we will revise our statements from “avoiding the vanishing gradient problem” to “expected to have more informative propagation during the backward pass”. For example, we will revise the statement “One-sided saturation avoids the vanishing gradient problem while maintaining noise-robustness.” as follows:
> >- “Activation functions that saturate only on one side are expected to have more informative propagation during the backward pass compared to those that saturate on both the positive and negative sides, by allowing for larger derivatives.”
>
> Clevert, Djork-Arné, Thomas Unterthiner, and Sepp Hochreiter. "Fast and Accurate Deep Network Learning by Exponential Linear Units (ELUS)." ICLR. 2016.
>
> Glorot, Xavier, Antoine Bordes, and Yoshua Bengio. "Deep sparse rectifier neural networks." JMLR workshop, 2011.
>
> Ramachandran, Prajit, Barret Zoph, and Quoc V. Le. "Searching for Activation Functions." ICLR workshop, 2018.
>
> ----
>
> ### Concluding remarks.
> - We sincerely appreciate your insightful comments. Your comments have significantly contributed to enhancing the quality of our presentation, especially on the statements about the properties of activation functions, and the noise-robustness of LayerAct and relationship between LayerAct and BatchNorm. If there are any further concerns or questions, we are more than willing to address them.

---

### Official Review · Reviewer_wLqf · 2023-11-01

**Soundness:** 3 good
**Presentation:** 3 good
**Contribution:** 3 good
**Rating:** 6
**Confidence:** 3

**Summary:**

The paper proposes a new activation function for CNNs with BatchNorm. The proposed layer-level activation, LayerAct, is designed to be more robust to noise and activation fluctuations due to shifts in input, compared to existing point-wise activation functions. The analysis and experimental results validate the noise-robustness of LayerAct.

**Strengths:**

- The paper presents a new layer-level activation function for CNNs with BatchNorm.
- The proposed activation function is presented in a general form in that variations of activation functions can be explored.
- The paper is well-written and easy to follow.

**Weaknesses:**

- The paper demonstrates experiments with rather same networks (ResNet) and small networks. Is the experimental result consistent with larger networks (e.g., ResNet-152) and different networks (e.g., EfficientNet, ResNext)?

- If LayerAct provides different effects from LayerNorm, there may be benefits the proposed function may bring for networks with LayerNorm. How does the proposed activation function behave with Transformers?

**Questions:**

Please refer to the weaknesses section.

---

> ### Author Response · Authors · 2023-11-14
>
> Thank you very much for your comments. We are pleased with your positive comments on LayerAct, particularly its capacity to overcome the limitations of element-level activation. In response to your concerns and questions, we describe our responses as follows.
>
> ### Weakness 1. Is the experimental result consistent with larger networks (e.g., ResNet-152) and different networks (e.g., EfficientNet, ResNext)?
> - We agree that evaluating performance on deeper and various networks is important.
> - Currently, we are training ResNet101 and ResNext50-32x4d with LA-SiLU and ReLU. We selected ReLU as the baseline element-level activation function since ResNet50 with ReLU exhibited superior performance on ImageNet compared to other element-level activation functions.
> - We will promptly share the results upon completion of these experiments to further address your concerns and improve our work.
>
> ### Weakness 2. How does the proposed activation function behave when it is placed in networks with LayerNorm?
> - The distinctive effect of LayerAct, compared to LayerNorm, lies in its ability to process the mean and variance statistics from activation input to activation output. However, if LayerNorm is placed before the activation layer, LayerNorm already normalizes the mean and variance of the activation input, leaving little room for diverse output statistics across samples.
> - Specifically, consider a network using LayerNorm without an affine function (no gain and bias), and placing LayerNorm layers right before activation layers. In such a network, the normalization output of the LayerNorm layers serves as the activation input. As a result, the normalization output of a LayerNorm layer is $n^{ \\hat{ LN } } = \\frac{ y -\\mu^2_y }{\\sqrt{\\sigma^2_y + \\alpha}}$. This leads to the output of SiLU $n^{ \\hat{ LN } } s\\left(n^{ \\hat{ LN } } \\right)$ and LA-SiLU $n^{ \\hat{ LN } } s \\left( n \\right)$ to be exactly same as $n = n^{\\hat{ LN }}_i$.
> - As such, the closer the output of the LayerNorm layer approximates the normalized input of LayerAct’s activation scale function $s$ (i.e., the more gain and bias approximate to one and zero, respectively), the more the benefits of LayerAct are reduced.
> - Please note that LayerAct has benefits as an activation: i) it addresses the trade-off problem between important properties of activation, and ii) has lower variance of noise-robustness across samples. These benefits can be maintained when used with LayerNorm that includes an affine function. This is because, in such scenarios, the layer-direction normalization in LayerNorm and LayerAct differs, represented as $n^{LN} = g \\cdot \\frac{ y -\\mu^2_y }{\\sqrt{\\sigma^2_y + \\alpha}} + b $ and $ n= \\frac{ y -\\mu^2_y }{\\sqrt{\\sigma^2_y + \\alpha}}$, respectively.
> - To validate this, we are conducting an additional experiment to train networks with LayerAct and LayerNorm on CIFAR10 and CIFAR100. We will report the outcomes of the experiments as soon as they are completed.
> - Based on your comment, we recognize that our manuscript currently lacks a detailed discussion on the potential limitations of LayerAct. We will incorporate a detailed discussion of these aspects in the Appendix, and ensure to reference this discussion in the main body of the manuscript. Thank you for your comment.
>
> ### Concluding remarks
> - We would like to reemphasize the main contribution of LayerAct: its capability to overcome the limitations of element-level activation functions. LayerAct can address the trade-off problem between important properties of activation and has smaller variance of noise-robustness across samples compared to element-level activation. Notably, our experimental results from image classification tasks demonstrate LayerAct’s capability to enhance the performance of ResNets on both clean and noisy datasets.
> - Once again, we sincerely appreciate your time and effort in reviewing our works. We could conduct additional experiments that can enhance our confidence in the performance of LayerAct, and improve the presentation of our manuscript. If there are any further concerns or questions, we are more than willing to address them.

---

> > ### Comment · Reviewer_wLqf · 2023-11-21
> >
> > I appreciate the authors' detailed responses. As for the weakness 2, if LayerAct has similar effects with LayerNorm, then I believe the proposed activation function may replace Layer Norm + activation function in existing transformer models. I suggest that authors take a look at possible benefits this additional experiment will bring to the final version of the paper.
> > As of now, my concerns are not addressed much without the experimental results available.

---

> ### Author Response · Authors · 2023-11-22
>
> We appreciate your second comment on the potential of LayerAct. Exploring the application of LayerAct functions in various architectures, such as transformer, is a key direction for our future research.
>
> ----
>
> ### Future research: transformers with LayerAct.
> - Our analysis of the relationship between LayerAct and LayerNorm, particularly the observation that LayerAct may offer reduced benefits with LayerNorm compared to BatchNorm, is confirmed by additional experiments on ResNets with LayerNorm, detailed in our Meta response 2-2.
> - Consequently, we are considering utilizing PowerNorm as the normalization method for the transformer with LayerAct functions, which is a variant of BatchNorm for transformer architecture in NLP tasks. We look forward to the opportunity to present these findings to you in the future.
>
> ----
>
> ### UNet and Unet++ with LayerAct.
> - Regrettably, due to limitations in computing resources, experiments on ImageNet with deeper ResNet or ResNext are not expected to be completed within the desired timeframe.
> - However, to demonstrate LayerAct’s potential for application in other architectures, we have reported the experimental results of UNet and UNet++ in Meta response 3.
> - In the experiment, networks with LA-SiLU outperformed those with ReLU and SiLU. It is noteworthy that the architectures of UNet and UNet++ differ from that of ResNet and are used for a different task, image segmentation.
>
> ----
>
> Once again, we appreciate your comments that helped us to set our future research with more confidence. We trust that the experimental results on image segmentation task may, at least partially, address your concerns regarding the consistent performance of LayerAct across various architectures.

---

> > ### Comment · Reviewer_wLqf · 2023-11-23
> >
> > Thanks for the detailed response. I have no further questions.

---

> > > ### Author Response · Authors · 2023-11-23
> > >
> > > We sincerely appreciate the time and effort you have dedicated to reviewing our paper. Your insightful comments have not only enhanced the quality of our work but have also helped define the direction of our future research.
> > >
> > > We are currently revising our manuscript and will make certain to incorporate responses to your review. Once again, thank you very much for your valuable feedbacks on our paper.

---

### Author Response · Authors · 2023-11-14
**Meta response**

- We appreciate the reviewers for their insightful feedback. Their comments have greatly aided us in clarifying the contribution and limitations of LayerAct, enhancing the presentation of our manuscript, and conducting additional experiments that augment our research.
- In this response, we aim to offer comprehensive meta-responses on the common and key questions from the reviewers. We trust this will help in providing a deeper understanding of our work. We will continue to update and add responses as needed.

---

> ### Author Response · Authors · 2023-11-21
>
> ## Meta response 1. Can LayerAct maintains its noise-robustness without BatchNorm?
>
> ----
>
> ### 1-1. Additional experiments on ResNets without a normalization method.
> - To verify whether the noise-robustness of LayerAct relies on BatchNorm, we conducted additional experiments on ResNets without a normalization method.
> - For these experiments, we selected ReLU and SiLU as baseline activation functions for comparison with LA-SiLU. The choice of ReLU was based on its simplicity as an activation function, while SiLU was chosen due to its use of sigmoid activation scale functions, similar to LA-SiLU.
> - We used the same experimental setting as in the experiments in our manuscript. We present the average accuracy over 10 trials.
>
> | Data | Model | Activation | Clean | Total | Noise | Blur | Digital | Weather | Extra |
> | - | - | - | - | - | - | - | - | - | - |
> | CIFAR10 | ResNet20 | ReLU |88.51 |67.43 |53.27 |62.12 |70.93 |75.91 |71.38 |
> | CIFAR10 | ResNet20 | LA-SiLU |88.95 |69.36 |54.71 |65.58 |71.93 |77.59 |73.34 |
> | CIFAR10 | ResNet32 | ReLU |89.56 |68.94 |54.48 |63.9 |72.45 |77.52 |72.73 |
> | CIFAR10 | ResNet32 | LA-SiLU |89.12 |70.59 |56.29 |66.81 |73.11 |78.94 |74.22 |
> | CIFAR10 | ResNet44 | ReLU |90.03 |69.97 |55.75 |65.08 |73.38 |78.45 |73.62 |
> | CIFAR10 | ResNet44 | LA-SiLU |88.77 |71.65 |58.15 |68.32 |74.01 |79.49 |74.88 |
>
> | Data | Model | Activation | Clean | Total | Noise | Blur | Digital | Weather | Extra |
> | - | - | - | - | - | - | - | - | - | - |
> | CIFAR100 | ResNet20 | ReLU |59.46 |37.69 |21.17 |38.76 |41.36 |44.15 |38.89 |
> | CIFAR100 | ResNet20 | LA-SiLU |61.01 |42.44 |26.07 |44.5 |45.74 |48.8 |42.98 |
> | CIFAR100 | ResNet32 | ReLU |60.71 |39.56 |23.05 |40.78 |43.23 |45.93 |40.67 |
> | CIFAR100 | ResNet32 | LA-SiLU |60.08 |44.01 |28.77 |46.73 |47.02 |49.51 |44.19 |
> | CIFAR100 | ResNet44 | ReLU |61.59 |40.88 |24.57 |42.08 |44.6 |47.16 |41.9 |
> | CIFAR100 | ResNet44 | LA-SiLU |60.3 |44.26 |29.22 |46.74 |47.45 |49.65 |44.5 |
>
> - The table presented above demonstrates the performance of ResNets without any normalization on both clean and noisy CIFAR10 and CIFAR100. We used a learning rate of 0.01. We report the average classification accuracy among 30 trials. ResNet32 and ResNet44 with SiLU exploded during training in some trials. Thus, we present the experimental results of the networks with ReLU and LA-SiLU.
> - The networks with LA-SiLU outperformed the networks with ReLU on noisy datasets. It is remarkable that networks with LA-SiLU demonstrated solid noise-robustness.
> - These experiments, along with subsequent experiments on ResNets with LayerNorm discussed in the following meta response, support that the noise-robustness of LayerAct functions is independent with BatchNorm.
>
> ----
>
> ### 1-2. Why networks with LayerAct perform poor on clean dataset?
> - LayerAct utilizes layer-direction normalization in its mechanism, but the output is not normalized. Therefore, it is more complex than element-level activation functions. This increased complexity could potentially lead to overfitting.
> - To empirically investigate this possibility, we conducted inference experiments on the CIFAR100 **training dataset** using networks trained without a normalization method.
>
> | data | model | ReLU | LA-SiLU |
> | - | - | - | - |
> | CIFAR100 | ResNet32 | 92.23 | 94.52 |
> | CIFAR100 | ResNet44 | 95.76 | 96.47 |
>
> - The experimental results show that networks with LA-SiLU have higher accuracy on the training dataset compared to those with ReLU and SiLU, which indicates overfitting.
> - The results from the shallower ResNet20 on the test dataset are consistent with this observation, as deeper networks are more prone to overfitting. These results support the idea that overfitting may be the main cause for the different performance of LA-SiLU in networks without a normalization method compared to those with BatchNorm.

---

> ### Author Response · Authors · 2023-11-21
>
> ## Meta response 2. LayerAct and LayerNorm.
>
> ----
>
> - Thanks to the reviewers, we recognized the lack of detailed discussion on the relationship between LayerAct and LayerNorm in the submitted manuscript. In response to reviewer RDYZ’s comments, we have noted that the benefits of LayerAct could be diminished when LayerNorm is employed.
> - This is due to the pre-normalization effect of LayerNorm before activation, which results in the activation output of a LayerAct function being similar to that of the corresponding element-level activation function. For details, we present our response to reviewer RDYZ and the results of additional experiments on ResNets with LayerNorm in the following sections: “Relationship between LayerAct and LayerNorm” and “Additional Experiments on Networks with LayerNorm,” respectively.
>
> ----
>
> ### 2-1. Relationship between LayerAct and LayerNorm.
> - When LayerNorm is placed before the activation layer, it normalizes the mean and variance of the activation input. This pre-normalization results in the activation output of a LayerAct function being similar to that of the corresponding element-level activation function.
> - For example, consider a network using LayerNorm without an affine function (no gain and bias) and placing the LayerNorm layer right before the activation layer. Here, the normalization output of the LayerNorm layers serves as the activation input. Consequently, in this scenario, the normalization output of a LayerNorm layer is $n^{ \\hat{ LN } } = \\frac{ y -\\mu^2_y }{\\sqrt{\\sigma^2_y + \\alpha}}$. This leads to the output of SiLU $n^{ \\hat{ LN } } s\\left(n^{ \\hat{ LN } } \\right)$ and LA-SiLU $n^{ \\hat{ LN } } s \\left( n \\right)$ to be exactly same as $n = n^{\\hat{ LN }}_i$.
> - As such, the closer the output of the LayerNorm layer approximates the normalized input of LayerAct’s activation scale function $s$ (i.e., the more gain and bias approximate to one and zero, respectively), the more the benefits of LayerAct are reduced.
> - Please note that LayerAct has benefits as an activation method, (LayerAct is not a normalization method): i) it addresses the trade-off problem between important properties of activation, and ii) has lower variance of noise-robustness across samples. These benefits can be maintained when used with LayerNorm that includes an affine function. In such scenarios, the layer-direction normalization in LayerNorm and LayerAct differs, represented as $n^{LN} = g \\cdot \\frac{ y -\\mu^2_y }{\\sqrt{\\sigma^2_y + \\alpha}} + b $ and $ n= \\frac{ y -\\mu^2_y }{\\sqrt{\\sigma^2_y + \\alpha}}$, respectively.
> - This implies that LayerAct can maintain its benefit when utilized with LayerNorm with an affine function, although it might be comparatively less than that on CNN-based networks with BatchNorm.

---

> ### Author Response · Authors · 2023-11-21
>
> ## Meta response 2. LayerAct and LayerNorm. (cont.)
>
> ----
>
> ### 2-2. Additional experiments on ResNets with LayerNorm.
> - To confirm our claims regarding the relationship between LayerNorm and LayerAct, we carried out experiments on ResNets with LayerNorm. We followed the same experimental setting of the experiments in our manuscripts. We present the average accuracy over 10 trials.
>
> - ResNet20 with LayerNorm on CIFAR10.
>
> | Data | Model | Activation | Clean | Total | Noise | Blur | Digital | Weather | Extra |
> | - | - | - | - | - | - | - | - | - | - |
> | CIFAR10 | ResNet20 | ReLU |88.24 |73.77 |62.05 |71.26 |76.48 |79.69 |76.46 |
> | CIFAR10 | ResNet20 | SiLU |87.53 |74.3 |63.32 |71.74 |77.33 |79.77 |76.58 |
> | CIFAR10 | ResNet20 | LA-SiLU |**88.52** |**75.31** |**63.81** |**73.11** |**78.23** |**80.92** |**77.6** |
>
> - ResNet32 with LayerNorm on CIFAR10.
>
> | Data | Model | Activation | Clean | Total | Noise | Blur | Digital | Weather | Extra |
> | - | - | - | - | - | - | - | - | - | - |
> | CIFAR10 | ResNet32 | ReLU |**88.55** |74.48 |63.33 |71.92 |76.94 |80.33 |77.11 |
> | CIFAR10 | ResNet32 | SiLU |87.27 |75.3 |64.78 |72.92 |78.55 |80.3 |77.32 |
> | CIFAR10 | ResNet32 | LA-SiLU |87.85 |**76.39** |**67.35** |**74.22** |**78.80** |**80.99** |**78.32** |
>
> - ResNet44 with LayerNorm on CIFAR10.
>
> | Data | Model | Activation | Clean | Total | Noise | Blur | Digital | Weather | Extra |
> | - | - | - | - | - | - | - | - | - | - |
> | CIFAR10 | ResNet44 | ReLU |**88.58** |75.2 |64.32 |72.75 |77.87 |**80.67** |77.67 |
> | CIFAR10 | ResNet44 | SiLU |86.65 |75.11 |65.67 |72.73 |77.84 |79.85 |77.1 |
> | CIFAR10 | ResNet44 | LA-SiLU |86.88 |**76.73** |**69.53** |**74.98** |**78.51** |80.43 |**78.43** |
>
> - ResNet20 with LayerNorm on CIFAR100.
>
> | Data | Model | Activation | Clean | Total | Noise | Blur | Digital | Weather | Extra |
> | - | - | - | - | - | - | - | - | - | - |
> | CIFAR100 | ResNet20 | ReLU |61.3 |44.04 |31.82 |44.39 |47.03 |48.86 |45.07 |
> | CIFAR100 | ResNet20 | SiLU |60.45 |**45.93** |**35.66** |**46.46** |48.46 |49.91 |**46.59** |
> | CIFAR100 | ResNet20 | LA-SiLU |**62.3** |45.3 |31.63 |45.9 |**48.72** |**50.65** |46.17 |
>
> - ResNet32 with LayerNorm on CIFAR100.
>
> | Data | Model | Activation | Clean | Total | Noise | Blur | Digital | Weather | Extra |
> | - | - | - | - | - | - | - | - | - | - |
> | CIFAR100 | ResNet32 | ReLU |**63.21** |45.92 |33.23 |46.17 |49.26 |50.94 |46.84 |
> | CIFAR100 | ResNet32 | SiLU |60.04 |47.14 |**37.37** |**48.6** |49.57 |50.33 |47.41 |
> | CIFAR100 | ResNet32 | LA-SiLU |60.76 |**47.26** |36.63 |48.51 |**49.97** |**51.03** |**47.51** |
>
> - ResNet44 with LayerNorm on CIFAR100.
>
> | Data | Model | Activation | Clean | Total | Noise | Blur | Digital | Weather | Extra |
> | - | - | - | - | - | - | - | - | - | - |
> | CIFAR100 | ResNet44 | ReLU |**64.18** |46.63 |33.92 |46.89 |**49.68** |**51.72** |**47.75** |
> | CIFAR100 | ResNet44 | SiLU |59.58 |**47.41** |**38.28** |48.69 |49.45 |50.61 |47.73 |
> | CIFAR100 | ResNet44 | LA-SiLU |60.17 |47.31 |38.1 |**48.82** |49.63 |50.26 |47.46 |
>
> - Our additional experiments on networks with LayerNorm substantiated our concerns about the relationship between LayerNorm and LayerAct. The results showed that ResNet32 and ResNet44 with ReLU outperformed those with LA-SiLU. On the noisy CIFAR100 dataset, the performance of ResNet20 and ResNet44 with SiLU was better than those with LA-SiLU.
> - Nonetheless, it is noteworthy that LA-SiLU exhibited superior performance in four out of six scenarios on noisy datasets. Moreover, networks with LA-SiLU outperformed those with SiLU on a clean dataset. Given that both LA-SiLU and SiLU utilize the sigmoid activation scale function, this suggests that the unique mechanism of LayerAct, which involves using layer-direction normalized values as input for the activation scale function, contributes to enhanced performance in networks with LayerNorm.
> - This observation brings us to the conclusion that the advantages of LayerAct are diminished when paired with LayerNorm as compared to BatchNorm. However, it’s noteworthy that the benefits of LayerAct might still be partially retained when LayerNorm is utilized, particularly in scenarios where the affine function is incorporated.

---

> ### Author Response · Authors · 2023-11-22
>
> ## Meta responses 3) LayerAct on UNet and UNet++ for an image segmentation task.
>
> ----
>
> - In our initial manuscript, we presented the experimental results exclusively for ResNets on image classification tasks.
> - We acknowledge the importance of validating LayerAct across various network architectures and tasks, as this can enhance confidence in the practical utility of LayerAct functions.
> - Here, we present the experimental results from UNet (Ronneberger et al., 2015) and UNet++ (Zhou et al., 2018) for a segmentation task on a nuclei image dataset. We selected BCEDiceLoss as the loss function for this experiment. We report the average IoU (Intersection over Union; %) over 10 trials with different weight initialization. DSV in the table denotes deep supervision (Lee et al., 2015).
>
> | Activation | UNet | UNet++ without DSV | UNet++ with DSV
> | - | - | - | - |
> | relu | 84.71 | 84.94 | 84.92 |
> | silu | 84.87 | 85.15 | 85.01 |
> | la_silu | **85.13** | **85.27** | **85.05** |
>
> - The experimental results demonstrate that networks with LA-SiLU outperform those with ReLU and SiLU in every case. This reveals the practical potential of LayerAct functions across CNN-based architectures and image segmentation tasks.
> - It is also important to highlight that neither UNet nor UNet++ utilizes a normalization method. Considering the experimental results from Meta response 1, where ResNet20 with LA-SiLU demonstrated better performance compared to that with ReLU and SiLU, this suggests that LayerAct can exhibit robust performance in shallow networks without normalization.
>
> Chen-Yu Lee, Saining Xie, Patrick Gallagher, Zhengyou Zhang, Zhuowen Tu. “Deeply-supervised nets.” PMLR. 2015.
>
> Ronneberger, O., Fischer, P., and Brox, T. “U-net: Convolutional networks for biomedical image segmentation.” MICCAI. 2015.
>
> Zhou, Zongwei, Md Mahfuzur Rahman Siddiquee, Nima Tajbakhsh, and Jianming Liang. "Unet++: A nested u-net architecture for medical image segmentation." DLMIA 2018.

---

### Author Response · Authors · 2023-11-23

## Revision of the manuscript.

----

- We appreciate the reviewers for their insightful comments. With such valuable feedback, we could improve the presentation of our manuscript, clarify our contribution with additional experimental results, and add discussion points that are important to our work.
- Based on the feedback, we have revised our manuscript. Changes include new content (colored in blue), moved sections (colored in brown), and shortened parts (colored in violet). Key revisions are summarized below:

----

### Clarifying the properties of activation functions.
#### Section 1 and Appendix A.
- To clarify the focus of our work on proposing a novel activation mechanism, we realized the necessity of a thorough and detailed discussion of existing activation functions.
- In the Introduction, we have clarified the properties and limitations of existing activation functions. Appendix A now contains an in-depth examination of the key properties of these activation functions.

----

### Clarifying our contribution is on advancing CNNs with BatchNorm with the novel activation mechanism.
#### Subsection 3.3, SubSection 4.2, Subsection 4.3, Appendix D, and Appendix E
- We clarified that our contribution is on advancing CNNs with BatchNorm, through discussions on the interplay of LayerAct with BatchNorm and LayerNorm.
- Experimental results indicate that the use of LayerAct can lead to overfitting more easily than element-level activations. This observation is corroborated by inference experiments of ResNets without normalization on training datasets.
- We clarified that the BatchNorm’s benefit of preventing the statistics of a channel from being similar across samples is crucial for the effective use of LayerAct.
- We clarified that the advantages of LayerAct may diminish when LayerNorm is placed before activation, as outputs may become similar to those of corresponding activation functions. This analysis suggests BatchNorm as a preferable option over LayerNorm.

----

### Clarifying the capability of LayerAct for enhancing the noise-robustness.
#### Subsection 3.2, Subsection 4.2, Appendix D, and Appendix E.
- We clarified that how LayerAct, our proposed activation mechanism, remains robust when inputs are not excessively large.
- To validate this, additional experiments on ResNets without normalization methods were conducted. We presented the result in Appendix E that demonstrate LayerAct's robustness independent of BatchNorm when inputs are not excessively large.
- We also included the experimental results of ResNets with LayerNorm and without normalization on noisy datasets in Appendix D and Appendix E, respectively. The results validate that networks with LA-SiLU exhibit enhanced robustness compared to those with ReLU and SiLU, affirming this advantage even in the absence of BatchNorm.

----

### Clarifying the relationship between LayerAct and normalization methods.
#### Subsection 3.3, Appendix D, Appendix E, and Appendix F
- We acknowledged that there was no discussion on the relationship between LayerAct and BatchNorm in our initial submission.
- The revised manuscript discusses that BatchNorm can improve the performance of LayerAct as: i) it can prevent overfitting due to the complex mechanism of LayerAct, ii) prevent the statistics of a channel to be similar across samples, and iii) prevent the input from being excessively large.
- In Subsection 3.3, we clarified how LayerAct's benefits may be reduced with LayerNorm due to pre-normalization effects. However, these benefits can be retained if LayerNorm incorporates an affine function, supported by additional experiments in Appendix D.
- We added additional experiments and analyses on ResNets with SwitchNorm, IEBN, and DBN in Appendix F. The outcomes of these experiments underscore the criticality of choosing an appropriate normalization method when utilizing LayerAct functions.

----

### Potential of LayerAct on different architectures and tasks.
#### Subsection 4.2 and Appendix I
- We have added experimental results for U-Net and UNet++ in Appendix I, showcasing LayerAct's potentials to be used in various architectures and tasks.

----

We sincerely appreciate the time and effort of the reviewers. Your insightful feedback has been instrumental in refining our work, enhancing the presentation, and conducting valuable additional experiments for improving the contribution. We hope that our revised manuscript aligns well with your expectations.

---

### Meta-Review · Area_Chair_4oif · 2023-12-06

**Metareview:**

The authors propose a novel activation mechanism for CNNs using BatchNorm. It is designed to be more noise-robust compared to existing element-level activation functions by reducing the layer-level fluctuation of the activation outputs due to shift in inputs. The authors present some empirical evidence that the proposed method indeed behaves as advertised and improves over other element-level activations.

Presenting yet another activation function at ICLR requires impeccable empirical results and wide applicability of the proposed method. In this case, the AC feels that focusing on (small) CNNs with BN is too narrow of a scope to be relevant to a large audience at ICLR. Beyond the narrow application potential, the technical contribution is modest. If the method generalized to other settings, such as vision transformers, the potential impact would be much larger. However, the initial results presented during the discussion phase show that when coupled with more "modern" approaches such as LN this method doesn't seem to provide additional benefits.

**Justification For Why Not Higher Score:**

- Too narrow of a scope (CNNs with BN)
- Small models

**Justification For Why Not Lower Score:**

N/A

---

### Decision · Program_Chairs · 2024-01-16

Reject